



# An extended history of high-amplitude lake-level changes in tectonically active Lake Issyk-Kul (Kyrgyzstan), as revealed by high-resolution seismic reflection data

A. C. Gebhardt[1*], Lieven Naudts[2,3], Lies De Mol[3], Jan Klerkx[4], Kanatbek Abdrakhmatov[5], Edward R. Sobel[6], M. De Batist[3]

[1]Alfred Wegener Institute Helmholtz Centre for Polar and Marine Research, Van-Ronzelen-Str. 2, 27576 Bremerhaven, Germany

[2]Royal Belgian Institute of Natural Sciences – Operational Directorate Natural Environment (RBINS-OD Nature), 3de en 23ste Linieregimentsplein, B-8400 Ostend, Belgium

[3]Renard Centre of Marine Geology, Universiteit Gent, Krijgslaan 281 s8, B-9000 Gent, Belgium

[4]International Bureau for Environmental Studies (IBES), Rue Audrey Hepburn 9/13, B-1090 Bruxelles, Belgium

[5]Kyrgyz Institute of Seismology, Bishkek, Kyrgyzstan

[6]International Universität Potsdam, Institut für Erd- und Umweltwissenschaften, Karl-Liebknecht-Str. 24, 14476 Potsdam, Germany

*Correspondence to*: A. C. Gebhardt (catalina.gebhardt@awi.de)

**Abstract.** A total of 84 seismic profiles mainly from the western and eastern deltas of Lake Issyk-Kul were used to identify lake-level changes. Seven stratigraphic sequences were identified each containing a series of delta lobes that were formed during former lake-level stillstands. Lake-level has experienced at least four cycles of stepwise fall and rise of 400 m or more. These fluctuations were mainly caused by past changes in the atmospheric circulation pattern during the past. During periods of low lake levels, the Siberian High likely was strong, bringing dry air masses from the Mongolian steppe. The strong Siberian High blocked the mid-latitude Westerlies. During periods of high lake levels, the Siberian High must have been weaker or displaced, and the mid-latitude Westerlies could bring moister air masses from the Mediterranean and North Atlantic regions.



## 1 Introduction

In the marine environment, global eustatic sea-level curves are traditionally used to reconstruct the amount of water stored on continents and in the oceans (e.g., Haq et al., 1987; Fleming et al., 1998; Lambeck et al., 2014; Dutton et al., 2015). Sea level is a good measure for the volume of water stored as

ice for shorter time periods during which other factors such as tectonic subsidence, seafloor spreading, thermal expansion can be ignored. Henge, sea-level curves, can be used to reconstruct glacial/interglacial cycles on a global scale. During the Last Glacial Maximum, for example, the eustatic sea level was some 125 to 135 m lower than today (e.g., Fleming et al., 1998; Lambeck et al., 2014). Lake-level curves, in contrast, often store a more local signal that might or might not be controlled by

glacial/interglacial cycles. Many large lakes with large water bodies and large volumes of sediment infill are fed by extensive catchments, and hence provide a powerful tool for understanding paleoenvironmental and paleoclimate change not only on local but also on regional scale. Changes in lake level can be in the order of some meters, but also be much larger than those recorded in the marine environment. Large-scale lake-level changes of up to several hundreds of meters were observed in a

series of large lakes, such as Lake Tanganyika (Lezzar et al., 1996), Lake Malawi (Scholz, 2007; Lyons et al., 2015), Lake Van (Cukur et al., 2014), Lake Lisan (Machlus et al., 2000), Lake Peten-Itza (Anselmetti et al., 2006), Lake Bosumtwi (Scholz et al., 2002) and Lake Challa (Moernaut et al., 2010). Lake Van is a large lake basin located in eastern Anatolia, Turkey (Degens et al., 1984; Litt et al., 2009). During the past 600 ka, i.e. since its formation, its water level changed by as much as 600 m

(Cukur et al., 2014). While climate forcing was identified as the dominant factor in driving lake-level changes in Lake Van, other factors such as volcanic and tectonic forcing could also be observed (Cukur et al., 2014; Stockhecke et al., 2014). Lake Petén Itzá is located in the lowland Neotropics of northern Guatemala on the Yucatan Peninsula. A paleoshoreline was identified at 56 m below present lake level, which means a reduction of ca. 87% of the total water volume at that time (Anselmetti et al., 2006).

Lake-level changes in Lake Petén Itzá were correlated to glacial/interglacial paleoclimate patterns with lowstands during the glacial and highstands during the interglacial periods (Anselmetti et al., 2006). Lake Lisan, the late Pleistocene precursor of the Dead Sea, is located in Israel. The ancient lake is not filled with water anymore, but its sediments crop out at several locations around the Dead Sea. Lake



Lisan existed between ~70 and 17 ka when it started to recede to the present-day Dead Sea lake level (Schramm et al., 2000). Lake-level changes are as large as 170 m (Bartov et al., 2002). Lake-level modulations in Lake Lisan have on one hand been attributed to paleoclimate change, but on the other hand, basin morphology and barriers between subbasins are able to modify and restrict lake-level changes in this area (Bartov et al., 2002, and references therein). Lake Bosumtwi is located in Ghana, West Africa. The impact crater lake is rather small with a diameter of ~8 km (Scholz et al., 2002). This lake is hydrologically closed (Shanahan et al., 2006). Lake level in Lake Bosumtwi, therefore, is purely driven by precipitation and the lake, hence, is sensitive on changes in regional (and global) climate.

Lake Issyk-Kul is a large lake located in Kyrgyzstan, Central Asia, in a tectonically active region surrounded by the Tien Shan Mountains. It is comparable in size with other large lakes worldwide, and thus likely to archive changes of the atmospheric circulation as well as indications of tectonic changes affecting the lake's drainage basin in its sediments. We use high-resolution sparker seismic data to reconstruct past water-level changes in Lake Issyk-Kul. Possible mechanisms are discussed that led to lake-level changes of up to 400 m, and the potential of the lake to help unravel regional paleoclimate change is shown.

## 2 Study area

### 2.1 Lake settings

Lake Issyk-Kul is an endorheic lake located in an intermontane basin in the northern part of the Tien Shan Mountains in Kyrgyzstan, Central Asia (42°30' N and 77°10' E, 1607 m altitude) between the relatively rigid Tarim Basin to the south and the Kazakh Platform to the north. The lake is bordered by the high mountains of the Terskei Alatau Range to the north (max. height 5212 m) and the Kungei Alatau Range to the south (max. height 4771 m) (Fig. 1). The lake is elongated: ~180 km E-W and ~60 km S-N. With a surface area of 6232 km$^2$, Lake Issyk-Kul is the second largest lake in the higher altitudes (De Batist et al., 2002). It has a mean water depth of ca. 278 m and an approximate water volume of 1,736 km$^3$ (Korotaev, 1967; Kodyaev, 1973; Zabirov, 1978).



The lake has a deep central basin with a flat bottom (668 m water depth) that extends over approximately 25% of the present lake area. Two large-scale shallow platforms characterize the lake at its western and eastern end, with the deltaic area being as wide as 60 km in the eastern and 40 km in the western part, and by steep slopes at the northern and southern shore with only a rather narrow shallower

shelf area. At the delta areas, the shelf is divided into two parts, one shallower part with water depths down to 110 m with an average inclination of 0.5°, and the other with water depths between 110 and 300 m and an average inclination of 1° (De Mol, 2006). Incised channels of up to 2-3 km width and 50 m depth are visible on both the eastern and the western shelf (Fig. 1), but are limited to the shallower part of the shelf. They are found in the prolongation of modern river mouths at the eastern part of the

lake, and are quite likely connected to former in- and outlets of the Chu river at the western delta (De Mol, 2006). The deeper part of the shelf is characterized by a series of terraces that were interpreted as ancient delta lobes, indicating lower water levels (De Batist et al., 2002).

The lake is fed by a total of 118 rivers and creeks draining an area of 22,080 km². These rivers mainly carry meltwater of snow and glaciers, rain and groundwater (Aizen et al., 1995). The largest rivers are

the Djyrgalan and Tyup rivers that feed into Lake Issyk-Kul at its eastern end. At present, Lake Issyk-Kul has no outlet, but during its history, it drained through the Chu River at its western end (De Batist et al., 2002). Approximately 640 km² of the drainage area are currently covered by glaciers that are located in altitudes of at least 3000 m asl. Most of these glaciers are found on the north flanks of the Terskei Alatau range. During the last glacial period, the glaciers extended down to the coast of Lake

Issyk-Kul (Grosswald et al., 1994).

Lake Issyk-Kul is oligotrophic to ultra-oligotrophic and well oxygenated though the entire water column down to the lake bottom. Surface water does not drop below 2-3°C in the winter and reaches values of 19-20°C during the summer. The lake is located in an arid area with deserts in the west, followed by semi-deserts and steppe towards east (Merkel and Kulenbekov, 2012). Salinity of the lake

water is currently approximately 6 mg/l (Merkel and Kulenbekov, 2012), but rising due to the lake currently functioning as a closed system.



## 2.2 Tectonic setting

The Tien Shan Mountains are one of the most important intracontinental orogenetic regions in Central Asia. Uplift and exhumation of the crystalline basement with its Paleozoic sediment cover has possibly started in late Oligocene times due to the progressive convergence of India and Eurasia after their collision in the Eocene (e.g., Goryachev, 1959; Molnar and Tapponier, 1975; Trofimov, 1990; Abdrakhmatov et al., 2002). During the Cenozoic, several strike-slip faults were active in this area, resulting in a transpressional tectonic context (Vermeesch et al., 2004). Exhumation of the Terskei Alatau and deposition of the basin fill began in the Late Oligocene (Macaulay et al., 2013; Wack et al., 2014; Macaulay et al., 2015). GPS measurements show that the Tarim Basin moves towards the north with approximately 15 to 20 mm per year with respect to Eurasia. The area is tectonically highly active as documented by recent and historic high-magnitude earthquakes (e. g. 1911: M 8.2) that often result in large subaerial landslides and quite likely also trigger large subaquatic mass movements. Most of the present-day tectonic activity is focused along the margins of the intermontane basins. Uplifted Pliocene lacustrine deposits, a clear example of the geodynamic activity in this area, are exposed at the southern shore of the lake where they are truncated by horizontal Quaternary lacustrine terraces, and some of the sediments were recently identified as deposited or deformed during earthquakes (Bowman et al., 2004).

## 3   Data acquisition and processing

First seismic data of the Lake Issyk-Kul sedimentary infill and architecture were acquired in 1982 by the Moscow State University with a total of 31 profiles across the lake. Unfortunately, only a few profiles were ever published (Stavinsky et al., 1984). Additional seismic profiles were acquired in 1997 and 2001 by the Renard Centre of Marine Geology (RCMG, Ghent University, Belgium). In 1997, 62 profiles (~990 km) were collected using a CENTIPEDE multi-electrode sparker (150-1500 Hz, operated at 500 J) as acoustic source (Imbo, 1998). In 2001, 40 additional profiles (~600 km) were acquired with the same source, and another 12 profiles using a SIG sparker (200-800 Hz, operated at 500 J). During both surveys, a single-channel streamer (2.7 m length, 10 hydrophones at 0.3 m spacing) was used as



seismic receiver (Naudts, 2002). Recording time and shot interval were chosen depending on the water depth of the specific area. The incoming signal was bandpass-filtered between 100 and 3000 Hz and subsequently digitized using a Triton-Elics DELPH2 data acquisition system with a sampling frequency of 5 kHz. Data were later converted to standard SEG-Y format for further processing. Navigation was

recorded using a SIMRAD Shipmate GPS system. Seismic processing comprised filtering, deconvolution, migration and amplitude scaling.

For the interpretation of the seismic database, all profiles were imported into KingdomSuite. Prominent sequence boundaries, both erosive and non-erosive, were mapped throughout all profiles except where either the sequences were not imaged due to the geographical location of the profiles or due to limits in

acoustic penetration/masking by the multiple. Even though sequence boundaries could not be mapped continuously between the eastern and western stacked-delta complexes, it was still possible to identify the same sequences in both areas due to similar two-way traveltime depths of the individual corresponding delta lobes.

Two-way traveltime was converted to depth below lake level (bll) using a sound velocity of 1500 m s$^{-1}$.

## 15  4   Results and interpretation

### 4.1 Tectonic origin of the lake

Most of the 84 sparker profiles used in this study are located either in the eastern or western shallow parts of the lake (Fig. 1). Only profile ik01 crosses the lake completely in the N-S direction. All profiles show that the sediments are well stratified in a layer-cake manner in the central part of the lake. On

profile ik01, the tectonic nature of the lake becomes obvious: Sediments dip towards the south, pointing to higher subsidence rates in that region (Fig. 2). This is visible to at least 200 m below the lake floor; in some places, these layers are even recognizable at a sediment depth of ca. 375 m. Within the imaged lake profiles, the faults responsible for the asymmetry of the basin are not visible, but studies along the southern margin of the lake document the presence of a series of faults roughly parallel to the long axis

of the lake (e.g., Burgette, 2008; Macaulay et al., 2014; Macaulay et al., 2015). The north-vergent Main Terskey Fault (MTF) is the most important basin-bounding fault, with over 8 km of structural relief



with respect to the lake (Macaulay et al., 2013). Apatite fission track studies show that this structure becaoem active in the latest Oligocene-Early MIocene. In the Late Cenozoic, the MTF propagated northward into the Issyk Kul basin; therefore, the anticline within the lake is likely linked to this structure.    In seismic profile ik01, the dip angle the strata seems to change quite continuously over time with no sign of an abrupt change, and the deformation seems to be still active, as the uppermost layers still display a slight dip angle. This points at an ongoing process. Short cores retrieved from the northern slope reveal that the sediment consists of a mixture of a terrigenous fraction (ranging from coarse sands to silty clays and clays) and a lacustrine micritic carbonate fraction. Reported sedimentation rates vary between 0.47-0.56 mm/yr (Ricketts et al., 2001), 0.49-0.59 mm/yr (Giralt et al., 2004), and 0.23-0.39 mm/yr (Larrasoaña et al., 2011; Gómez-Paccard et al., 2012) based on [14]C and nuclide dating. Using a mean value of 0.45 mm/yr, this points at a minimum age of ca. 830 ka for the lowermost sediment layers visible in the profiles. However, it is however quite likely that the lake is significantly older, because the acoustic basement was not detected in any of the profiles. Furthermore, the sedimentation rates used here are derived from cores located proximal to the northern shore, and sedimentation rates might be significantly lower in the central part of the lake. Deep bore holes and outcrops within the basin adjacent to the lake reveal up to 5 km of Cenozoic strata (Knauf, 1965; Turchinskiy, 1970; Fortuna, 1983); the older lacustrine record is poorly studied.

In the southeastern part of the lake, the strata are not inclined as would be expected in this asymmetric basin. This is due to two anticline structures that are visible over a length of ca. 30 km and 10 km, respectively (Fig. 1). The longer of these two is only observed in the deeper parts of the lake and its prolongation further towards east is likely masked by the stacked delta sequences that strongly reduce penetration of acoustic energy. The second anticline structure seems to be parallel to the first one, but located some 8 km further south. Both anticlines are progressively buried by younger sediments, and the southern one is meanwhile completely leveled by sediments.



## 4.2 Facies types

While the central part of the lake is characterized by inclined strata that were deposited in a layer-cake manner, the slope and shelf are much more diverse in their acoustic image. We identified four different facies types (Fig. 3) that can be described as follows:

*Facies I: Delta lobes with topsets and foresets.* This facies type is characterized by a succession of prograding clinoforms (Fig. 3a). The reflectors have variable amplitudes with generally lower amplitudes in the deeper sequences. This is rather the result of limited acoustic penetration than a real signal. This facies type is found throughout almost all stratigraphic sequences both in the eastern and

10 western delta area. Facies I is interpreted as prograding delta lobes with the characteristic lateral and/or vertical succession of topsets, foresets and bottomsets. The sediments likely consist of coarse-grained material brought into the lake by the large rivers such as the Tyup and Djyrgalan rivers in the eastern and the Chu River in the western delta.

15 *Facies II: Prodelta sediments.* This facies type is characterized by parallel reflectors of high amplitudes that can be followed over distances of several kilometers (Fig. 3b). The reflectors form a drape or onlap onto the underlying reflectors, and they are mostly found in locations distal to Facies type I. This facies is interpreted as distal deltaic sediments, i.e. prodelta sediments, and likely consist of fine-grained material brought by the rivers, possibly intercalated with turbidites.

*Facies III: Transgressional deltaic sediments.* This facies type is similar to the topset part of facies type I, but it consists of several horizontal layers that get successively closer to the coastline during their deposition (Fig. 3c). It forms onlaps onto the underlying sequence boundary or reflector. This facies type is interpreted as transgressional deltaic sediments during times of rapid lake-level rise.

*Facies IV: Acoustically transparent.* This facies type is acoustically transparent (Fig. 3d). Where amplitudes are high enough for the reflectors to be detected, they are chaotically distributed. This facies is mainly found in the deeper parts of the delta where it forms a thick package of sediments, laterally





bounded by sediments of facies type II towards the more distal part of the delta. Facies IV may be interpreted as former delta sediments that have been affected by post-depositional processes (e.g. sediment remobilization, slumping, liquefaction) that caused them to loose their internal structure.

### 4.3 Stratigraphic sequences

The seismic profiles across the eastern and western delta areas were interpreted in terms of stratigraphic sequences. Boundaries between the different sequences are often erosive, but in places they are also non-erosive. Erosive boundaries, i.e. erosional unconformities marked by truncation of underlying reflectors and/or irregular morphology, point at a lake-level fall while non-erosive boundaries, i.e. conformities, were formed during times with stagnating or rising lake levels.

Seven stratigraphic sequences were identified. The sequences are imaged on almost all seismic profiles except where they were masked by the lake-floor multiple, where acoustic penetration was limited, or where the location of the profiles was not suited to image that specific sequence. Sequences 3 and 4 were clearly identified on the western profiles, but could not be differentiated from each other on the profiles across the eastern delta area.

The sequences are described in the following sections following the stratigraphic order, i.e. from the deepest/oldest sequence (Sequence 7) towards present (Sequence 1).

We use delta depth (expressed in meters below current lake level) as an indicator for past lake-level change. The topset-foreset roll-over point is considered as a proxy for the lake level at the time of its forming. The topsets are deposited subaqueously at approximately 30 m below lake level (the current

topset-foreset roll-over point is at 28 m bll). No corrections were made for compaction and/or tectonic subsidence.

### 4.3.1 Sequence 7

Sequence 7 is the lowermost sequence that is visible in the seismic profiles (Fig. 5b). Its lower boundary is masked in places by the multiple and/or limited penetration of the acoustic signal. Where its lower

boundary is visible, it is clear that the seismic survey did not penetrate down to the acoustic basement, but that Sequence 7 is actually overlying (lacustrine) sediments of unknown thickness. Within Sequence





7, a series of 5 delta lobes could be identified, i.e. spatially well-defined parts of the sequence characterized by the seismic Facies type I. Some of these occur only in the western delta area (7.5, 7.4), some only the eastern delta area (7.2), and some in both areas (7.3, 7.1). The stratigraphic succession of delta lobes 7.1 to 7.5 indicates a stepwise lake-level fall with stillstands at 330, 381, 412, 454, and 504

5   m bll.

### 4.3.2 Sequence 6

Sequence 6 is clearly visible both on the western and eastern delta areas. It is overlying Sequence 7 and is rather thin exhibiting erosive upper and lower boundaries in many profiles (Fig. 5a). Nevertheless, two delta lobes could be identified at 461 (no. 6.1) and 361 m bll (6.2). Sequence 6 can thus be

interpreted as deposited during lake-level rise with a first stillstand at 461 and a second stillstand at 361 m bll.

### 4.3.3 Sequence 5

Sequence 5 is overlaying sequence 6 with an erosional boundary in between. It is characterized by two deltas at 284 m bll (no. 5.1) and 364 m bll (no. 5.2). The bathymetrically higher delta 5.1 exhibits

extensive erosion (Fig. 6b). The bathymetrically lower delta 5.2 is still visible in the modern lake floor morphology as it is only draped by the overlaying Sequence 4; it forms the current shelf edge. Sequence 5 can therefore be interpreted as having formed during a step-wise lake-level fall from a first stillstand at 284 m bll to a second stillstand at 364. It is quite likely that extensive erosion of the upper part of delta lobe 5.1 took place during the deposition of the lower lobe 5.2.

### 4.3.4 Sequence 4

Sequence 4 is overlies sequence 5, and it is visible in both the western and eastern delta areas. On the latter, its upper boundary with Sequence 3 is unclear due to the fact that Sequence 3 is not clearly visible in this area. Sequence 4 in the western delta area either includes Sequence 3 and the boundary in between is not visible, or Sequence 3 is completely eroded here. In Sequence 4, the delta lobes are

characterized by mainly acoustically transparent sediments of Facies type IV, but the sequence also contains packages of well-layered prodeltaic sediments of Facies type II (Fig. 6a) in the distal parts




distal of the different delta lobes. Three delta lobes were identified: the oldest (4.1) at ca. 319 m bll, followed by a delta (4.2) at approximately 250 m bll and a third, younger (4.3), at 397 m bll.

The interpretation of Sequence 4 indicates a lake-level rise from a lower stillstand at 319 m bll to a second stillstand at 250, followed by a subsequent lake-level fall with another stillstand at 397 m bll.

### 4.3.5 Sequence 3

Sequence 3 could only be clearly identified in the western delta areas; in the eastern area, Sequence 3 either cannot be distinguished from Sequence 4, or it is completely eroded (Fig. 6a). In the western delta complex, Sequence 3 is characterized by a lower boundary that was partially erosive into the underlying sediments but grades into a correlative conformity in other places. Two distinct delta lobes were identified within Sequence 3, the older at 330 (3.1) and the younger at 172 m bll (3.2; Table 1).

Sequence 3 can be interpreted having formed during a lake-level rise with two stillstands, one at 330 m bll, followed by a rapid lake-level rise and a second stillstand at 172 m bll.

### 4.3.6 Sequence 2

Sequence 2 is visible in almost all profiles in the eastern and western delta areas (Fig. 7b). In the eastern delta area, due to the partial acoustic transparency of the underlying Sequence 3 it is not clear if the lower boundary of Sequence 2 is erosive or non-erosive; where Sequence 3 is not acoustically transparent, but layered, the boundary between sequences 2 and 3 seems to be non-erosive. In the western delta area, the lower boundary of Sequence 2 is clearly non-erosive. Sequence 2 is characterized by 2 delta lobes that were formed at 210 (2.1) and at 250 m bll (2.2; Table 1).

Sequence 2 can be interpreted as a succession of step-wise, slow lake-level fall with stillstands at 210 and later at 250 m bll. During the lake-level fall, erosion may have taken place in the upper, proximal parts of the lake that were aerially exposed, but this is not visible in our seismic network.

### 4.3.7 Sequence 1

Sequence 1 could be identified on seismic profiles in both the eastern and western delta areas. It contains the uppermost, youngest sediments and its upper boundary forms the current lake floor (Fig. 7a). It lies above an erosional unconformity with channels deeply incised (up to 35 m) into Sequence 2



at several spots (Fig. 7a). Sediments of Sequence 1 fill the channels. In its distal part, it drapes the underlying topography with a shelf break at ca. 340 m and prodeltaic sediments of Facies type II deeper down. The lowermost part of Sequence 2 forms a small delta lobe (1.1) at approximately 285 m bll. This lobe is overlain by a succession of transgressional units of Facies type III. On some profiles, a

series of smaller delta lobes (1.2 to 1.5) is visible at water depths of 263, 251, 228, and 201 m bll (Table 1). A distinct large delta lobe (1.6) is visible in almost all profiles at a water depth of ca. 153 m bll. Only on profiles issyk024 and ik07 that reach into the shallowest parts of the lake on the eastern delta area, the uppermost, currently active delta lobe (1.7) at 28 m bll was identified. The distal prodeltaic sediments associated with delta lobe 1.7, however, can be identified as thin drape on almost all profiles

of both the eastern and western delta areas.

The lake-floor morphology shows a shelf break at approximately 150 m bll. The large shallow areas above the present-day delta at ca. 30 m bll are characterized by subaquatic channels that begin at the mouths of the large rivers at the eastern shore and in front of the paleo-channel of the Chu River at the western shore. These channels can be followed over the entire plateaus and end at approximately 110

bll.

Sequence 1 can be interpreted as starting with a relative lake-level lowstand during the formation of its lowermost delta 1.1. During this lake-level lowstand, erosion took place in the hinterland and likely formed the river incisions that are visible at the boundary between Sequences 1 and 2 on profile issyk019 (Fig. 7a) and in nearby profiles. The lake-level lowstand was followed by a rapid

transgressional phase ending with a slightly slower lake-level rise from 263 to 201 m bll. The sedimentary infill of the deeper channel likely was deposited during this transgressional phase. A second lake-level stillstand took place during the formation of the delta 1.6, followed by the current situation in which delta 1.7 is being deposited approximately 28 m bll.





## 5 Discussion

### 5.1 Lake-level curve and age information

Combining all information on lake-level stillstands, delta formation, regressional and erosional phases, a lake-level curve for Lake Issyk-Kul was established (Fig. 8). It comprises 4 phases of lake-level

decreases (in Sequence 7, Sequence 5, second part of Sequence 4, and Sequence 2) and 4 phases of lake-level rise (Sequence 6, first part of Sequence 4, Sequence 3, and Sequence 1). The transition between the different sequences and thus the transition between decreases and increases cannot be clearly described due to the fact that the boundaries are mostly of erosive nature, mainly in the upper parts of the lake. This is easily explained by the observation that the upper parts of the sediments must

have fallen dry during the formation of lowstand deltas and thus were subject to erosion.

The lake-level curve in its present form only comprises the deltas/terraces that were formed inside the current lake; subaerially exposed terraces are not imaged in the seismic data, and their distal counterparts, i.e. the distal well-layered prodelta sediments, are likely not identified because they lie concordantly on the underlying sediment.

Different authors describe a series of terraces that are today subaerially exposed. The uppermost terrace in the lake area is located at 1675-1680 m above sea level (asl) (Trofimov, 1990), which is some 70 m above the present-day lake level. Bowman et al. (2004) describe beach cliffs at an altitudinal range of 1620-1640 m asl. With the present-day delta being formed in a water depth of approximately 30 m, these subaerially exposed terraces point at lake levels that were 100 and 43-63 m higher than present.

Bowman et al. (2004) date these terraces to ages between 26.0 ± 2.1 ka for the upper ones and 10.5 ± 0.7 ka for the lower ones, which is in agreement with an older date of 26.34 ± 0.54 ka for an upper terrace (Markov, 1971). This means that the lake-level highstand at 63 m above lake level occurred roughly at the beginning of MIS2. Lake Issyk-Kul's lake surface today is almost at the spill-over level. Terraces some 63 and 100 m higher than the current lake surface would mean that the outflow must

have been dammed by a dam some 63 and 100 m higher than today for a substantial time so terraces could have formed. With the current topography, this scenario is rather unlikely, and the subaerial terraces might also just have been uplifted from their original position. On the other hand, young lacustrine sediments are interbedded with fluvial conglomerates in Boom gorge, suggesting that the



narrow gorge was dammed in the past. This could have blocked the current lake's spill-over point, maybe resulting in lake levels higher than would be possible today.

Subaquatic channels are visibly incised into Sequence 1 (Fig. 1) on the large shallow parts of the eastern stacked-delta complexes. On the eastern complex, these are associated with the prolongation of the rivers that currently feed Lake Issyk-Kul. On the western complex, the paleo-channel marks the ancient position of where the Chu River entered the lake before it was redirected (De Batist et al., 2002). Today the Chu River flows some 10 km west of the lake. The channels were likely formed at a lake-level lowstand at 110 m bll, which – taking into account the 30 m of water that are today measured ontop of the currently active delta – corresponds to a delta depth of roughly 140 m. The channel morphology is still very distinct, pointing at a rather young event during which they were formed, so it is quite likely that they eroded into the subaerially exposed sediments of Sequence 1 while delta 1.6 was formed at a water depth of 153 m. Older Russian literature dates this lowstand as following the MIS2 regression marked by the subaerial terraces mentioned above (Markov, 1971; Bondarev and Sevastyanov, 1991). This would imply that delta 1.6 was deposited after a significant lake-level fall of more than 100 m. Seismic data, however, show a succession of a delta (1.1) at a water depth of 285 m bll, a transgressional phase followed by a stepwise increase with small-scale deltas (1.2 to 1.5) at water depths of 263, 251, 228, and 201 m bll before the formation of delta 1.6 at 153 m bll, with no sign of a lake-level highstand in between. It is thus unlikely that lake level was high before the formation of delta 1.6, providing evidence that the subaerially exposed lake terraces might not be located in their original position but significantly uplifted.

### 5.2 Lake-level variations and their trigger mechanisms

Lake-level variations are always a sign of changes in the hydrological regime of a lake. Basically, two different mechanisms can affect the hydrological regime: (i) a change in lake geometry, e.g. (differential) subsidence, a blocking of the outlet stream or, contrarily, the formation of a new outlet, and (ii) a change in the balance between precipitation/inflow and evaporation/outflow.





### 5.2.1 Changes in lake geometry

Lake Issyk-Kul is located in a tectonically highly active area, which at first glance makes it likely that lake-level changes may have their primary origin in tectonic events. The lake basin is located in the intramontane Issyk-Kul Basin that is separated from the surrounding mountain ranges (the Kunghei

Alatau towards north and the Terskei Alatau towards south, Fig. 1) by fault zones. Most of the Late Cenozoic strain has been accommodated by the adjacent mountain ranges, and as a result, the Issyk-Kul Basin and the lake basin in its center have been mostly protected from strong deformation (Abdrakhmatov et al., 2002). This is confirmed by the mostly well-layered sediments observed in the lake; however, evidence for tectonic influence within the lake basin is present (e.g. the tilting of the

central basin deposits towards the south, the anticline structures in the east, and the occurrence of mass transport deposits likely caused by underwater slope destabilization due to seismic shaking).

Lake-level changes could be linked to tectonically-driven subsidence or exhumation that would (a) either influence the entire lake, (b) parts of it, or (c) the inlets and/or outlets.

(a) From the well-layered sediments in the central part of the lake it is quite obvious that the general

sedimentation pattern did not change significantly during the time interval which we can observe. The strata are almost perfectly horizontal in W-E direction, but dip gently towards south with dip angles increasing with depth. This points at differential subsidence only between the northern and the southern part of the lake, and the subsidence seems to have been relatively constant through time. In order to generate such a highly dynamic lake-level curve as for Lake Issyk-Kul (Fig. 8) with several cycles of

increases and decreases and a total difference of at least 400 m on a time scale of ~1 Ma, it is virtually impossible that the lake-level variations were generated by fault driven uplift or subsidence within the lake. A general trend is nonetheless probable and confirmed by the differential subsidence between the northern and southern shore imaged in the sediments, but superimposed by another mechanism that is responsible for the dynamic change in lake level.

(b) The two anticlinal structures visible in the eastern part of the lake may have influenced the lake at its eastern end, but given that the delta depths for almost all delta lobes are almost identical on both the eastern and western delta, it is unlikely that these two areas experienced significantly different histories. Even though some delta lobes were only identified on one delta, many other lobes are observed at



identical depths on both deltas. In the case of Sequence 7, lobe 7.1 was observed in 327 m bll in the west and 333 m bll in the east, lobe 7.2 is missing in the west, 7.3 was observed in 415 m bll in the west and 409 m bll in the east, lobe 7.4 was found at 454 m and lobe 7.5 at 504 m bll in the west only. The differences between east and west are not larger than those between the lobe depths identified within

one delta if identified in several profiles. Also, for example lobe 7.5 was only identified on one profile in the western delta. It is therefore likely that those delta lobes that were only observed in one of the deltas are in fact not missing in the other, but were just not identified in our study, either because our seismic profiles were not placed perfectly to image that specific lobe, or maybe even because this lobe was eroded after its deposition.

(c) Tectonic events can also influence inlets and outlets of a lake, which in turn at least partly control lake level. At present, Lake Issyk-Kul is a closed system without any outlet, but in former times, the lake was drained through an outlet at its western end. The Chu River entered the lake in the southern and left it in the northern part of the western delta. At some point during the late Pleistocene, its course changed and it currently bypasses the lake and flows through Boom Canon towards northwest due to a

gentle topographic barrier between the river and the lake. This barrier was likely caused by changes in the tectonic settings (Bondarev and Sevastyanov, 1991). This means that Lake Issyk-Kul not only has no current outlet, but also no large inflow through the western delta. Should the Chu River keep its current course, this would result in a decline in sediment accumulation in the western delta, and delta lobes would likely not develop as pronounced in future in this part of the lake. A lake-level rise of 13 m,

however, would flood the barrier and reactivate the Chu River as an outflow. In the beginning of the 19[th] century, Lake Issyk-Kul drained through the Chu River for some 25 years (Merkel and Kulenbekov, 2012). The maximum documented lake level that Lake Issyk-Kul has experienced is at 1,680 m above sea level (asl), i.e. 73 m above present lake level (Trofimov, 1990). With the current topography, it would not be possible to generate terraces at 73 m above lake level, as the lake would

overflow at +13 m already. This suggests that during the period where water reached to 73 m above lake level, either the current possible overflow channel – the Boom Canyon - was blocked, or the lake surroundings have been uplifted relative to the lake itself since that time and the +73 m terraces were initially deposited at another – lower – altitude. Or the entire lake basin and lake have been uplifted and



all terraces are not at their original position. As the present structural setting would be predicted to raise the range with respect to the lake, the latter scenario seems unlikely. Grosswald et al. (1994) report that alpine glaciers have extended to the present shoreline and likely into the Chu river valley in the past, so they may have dammed the lake and facilitated lake levels higher than currently possible.

It is unlikely that the current bypassing of the Chu River is a common situation in the lake's history. In addition to the blocked outlet, this in fact also means that a large river entering from the west is missing. The pronounced delta lobes that formed contemporarily in the east and west point at sediment sources at both ends during at least the period that is spanned by sequences 7 to 1.

All things considered, we propose that the lake-level fluctuations identified on our seismic data do not

originate from tectonic activity only. Tectonic activity in this region might result in an overall trend of the lake level, but not in the highly dynamic, repeated cycles of fall and rise. These must have been the result of significant changes in precipitation and/or evaporation.

### 5.2.2 Changes in precipitation/evaporation

Today, climate and hence precipitation/evaporation in the Tien Shan is mainly controlled by the

interaction between the mid-latitude Westerlies and the Siberian Anticyclone (e.g., Aizen et al., 1997; Zech, 2012). The mid-latitude Westerlies bring moisture from the Aral-Caspian Basin, the Mediterranean, the Black Sea and the North Atlantic (Aizen et al., 2006; Lauterbach et al., 2014), while the Asian summer monsoon has been only of minor importance at least since the Mid Holocene (Cheng et al., 2012). The Siberian Anticyclone reaches south to the Tien Shan area and blocks the mid-latitude

Westerlies during winter, which results in low winter precipitation due to the dry air masses of the Siberian High (Aizen et al., 1995; Aizen et al., 2001; Ricketts et al., 2001). Maximum precipitation is observed in spring and summer, and additionally in autumn in areas with altitudes of less than 3000 m (Aizen et al., 2001). Between 1931 and 1990, precipitation has increased significantly in the northern Tien Shan by up to 108 mm (Aizen et al., 2001). Aizen et al. (2001) interpret this increase as a stronger

influence of the mid-latitude Westerlies on precipitation, which might result from an increase in global air temperature and a weakening or displacement of the Siberian High. Lake level, however, has fallen by 3 m since 1926 (Ricketts et al., 2001), partly due to Soviet era hydrological projects, but also by





increased evaporation due to increased temperature (Romanovsky, 1990; Ricketts et al., 2001; Romanovsky, 2002).

Lake Issyk-Kul is mainly fed by riverine input, which in turn is mainly controlled by snow and glacier melt. Lake-level changes are therefore highly dependant on precipitation and evaporation, which in turn

are controlled by moisture and air temperature. With lake-level changes of up to 400 m, the lake's archive shows that the interplay between Siberian High and mid-latitude Westerlies has been highly dynamic during the past, and likely since the formation of the lake. Unfortunately, we do not have age information from the delta lobes. Even though some age and sediment accumulation information is available from short sediment cores mostly from the mid-slope area (e.g., Ricketts et al., 2001; Giralt et

al., 2004), these data are not helpful to estimate ages of the different delta lobes. Delta systems are just too dynamic with large differences in accumulation rates depending on the proximity to the rivers and to their total sediment load (bedload and suspended load) to use generalized accumulation rates. Additionally, erosional contacts between the different sequences could be observed but the amount of missing sediment could not be quantified. Drilling of sediment cores from the different sediment lobes

would be required in order to reliably date the lake-level stillstands. Lake Issyk-Kul with its long sedimentary archive therefore provides an ideal drill site to study the history of the interplay between the Siberian Anticyclone, the mid-latitude Westerlies and the Asian summer monsoon through time (Oberhänsli and Molnar, 2012). The lake is located in an area with high relief, implying different climate conditions over short distance. Nevertheless, lake-level changes of up to 400 m are significant

and rather a result of large-scale changes in climate patterns rather than of regional differences.

In an arid area such as the Issyk-Kul region, glaciers are highly sensitive to changes in precipitation/evaporation. Evidence from glacial features such as moraines, however, is limited and does not reach further back in time than to MIS6 (e.g., Zech, 2012); in the Kyrgyz Tien Shan, existing ages only reach back to MIS5e (Koppes et al., 2008). Zech (2012) showed that the glaciation in the Tien

Shan and Pamir became successively more restricted from MIS4 to MIS2, which was also observed in Siberia with limited ice sheet and glacier extents (e.g., Svendsen et al., 2004; Zech et al., 2011). Zech (2012) interprets this as a result from reduced moisture advection through the mid-latitude Westerlies due to the lee effect when flowing over the massive Fennoscandian Ice Sheet, and additionally by a





blocking situation of the Westerlies by a strong Siberian High. Low lake level would likely be caused by low precipitation and, thus, points at a strong Siberian High. This would imply that sequence boundaries 7/6, 5/4, 4/3, 2/1 represent times of a strong Siberian High and weakened mid-latitude Westerlies. Sequence boundaries 6/5 and 3/2, in turn, would point at a displaced or weakened Siberian

High, with enhanced moisture input from the mid-latitude Westerlies and, possibly, also by the Asian summer monsoon. The extent of the past glaciations in the Arctic region was variable (e.g., Svendsen et al., 2004; Jakobsson et al., 2014), which makes it likely that the precipitation regime was also different. Glaciation in the Tien Shan might have been rather limited during arid glacials and lake level may have been low. During glacials with wetter conditions, glaciers may have reached the lake and may even

have blocked the outlet, potentially leading to the highest lake levels recorded, higher than during (moist) interglacials without blocked outlet. It is therefore impossible to relate sequence boundaries to glacials or interglacials without direct dating of sedimentary material.

## 6   Conclusions

A seismic study of mainly the western and eastern delta of Lake Issyk-Kul exhibits seven stratigraphic

sequences (Sequence 7 = oldest, Sequence 1 = youngest). Boundaries between the sequences are often erosive (erosional unconformities), and the stratigraphic sequences each contain at least 2 delta lobes formed during past lake-level stillstands. The topset-foreset roll-over point was considered as a proxy for lake-level change. In this context, Sequences 7, 5, and 2 can be interpreted as formed during falling lake level, while Sequences 6, 3 and 1 indicate a lake-level rise. Sequence 4 exhibits a lake-level rise

followed by a lake-level fall. Taking into account the subaerially exposed lake terraces, a total of at least 400 m of lake-level change could be observed.

While tectonic reasons may have had some influence on lake-level evolution, they were clearly not causal for the cyclic fall and rise. Changes in precipitation/evaporation are more likely causing changes in water level. Currently, the dry air masses of the Siberian High are strong in winter, blocking the mid-

latitude Westerlies. During summer, the mid-latitude Westerlies bring moist air into the area, resulting in precipitation peaks in spring/summer and autumn. Lake-level changes point at changes in the atmospheric circulation pattern during the past. Low lake levels point at less precipitation, likely with a




strong and stable Siberian High, and high lake levels may have been caused by a weakened or displaced Siberian High and stronger mid-latitude Westerlies, possibly even influence of the Asian summer monsoon.

## 7  Data availability

Seismic profiles are stored at Renard Centre of Marine Geology, Universiteit Gent, Belgium, and can be obtained from Marc De Batist upon request (Marc.DeBatist@UGent.be).

## 8  Acknowledgements

The seismic surveys were organized with support of the Belgian Science Policy Office (BELSPO) and of the EU FP5 APELIK project. We like to thank the captain and crew of RV Møltur for the support

during the seismic campaigns in 1997 and 2001.

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



| Lobe No. | Western delta | | | Eastern delta | | | | | | | Combined | |
|---|---|---|---|---|---|---|---|---|---|---|---|---|
| | issyk049 | ik30 | ik28 | issyk017 | issyk019 | issyk021 | ik49 | issyk022 | issyk024 | ik07 | mean | std dev |
| 1.7 | | | | | | | | | | | 28 | 2 |
| 1.6 | 161 | 155 | 155 | 154 | 151 | 147 | 154 | 153 | 149 | 154 | 153 | 4 |
| 1.5 | | 228 | | | | 199 | 203 | | | | 201 | 3 |
| 1.4 | | | | | | 228 | | | | | 228 | 0 |
| 1.3 | 252 | 246 | | | | 250 | 254 | 254 | | | 251 | 3 |
| 1.2 | | | | | | 263 | | | | | 263 | - |
| 1.1 | | | | 294 | 279 | 283 | | | | | 285 | 8 |
| 2.2 | 244 | | 244 | 252 | 248 | 261 | | | | | 250 | 7 |
| 2.1 | | 206 | | 215 | 203 | 217 | | | | | 210 | 7 |
| 3.2 | 174 | 180 | 162 | | | | | | | | 172 | 9 |
| 3.1 | | | 330 | | | | | | | | 330 | - |
| 4.3 | 399 | 395 | | | | | | | | | 397 | 3 |
| 4.2 | 252 | 255 | 251 | | | 241 | | | | | 250 | 6 |
| 4.1 | 315 | 327 | 314 | | | | | | | | 319 | 7 |
| 5.2 | 366 | 373 | 379 | 359 | 358 | 350 | | | | | 364 | 11 |
| 5.1 | 280 | 274 | | 283 | 295 | 286 | | | | | 284 | 8 |
| 6.2 | 386 | 371 | 370 | 335 | 344 | 360 | | | | | 361 | 19 |
| 6.1 | 474 | 448 | | | | | | | | | 461 | 18 |
| 7.5 | 504 | | | | | | | | | | 504 | - |
| 7.4 | 471 | 448 | 444 | 406 | 405 | 417 | | | | | 454 | 15 |
| 7.3 | 424 | 411 | 410 | 384 | 381 | 378 | | | | | 412 | 7 |
| 7.2 | | | | | | | | | | | 381 | 3 |
| 7.1 | 326 | 320 | 335 | 330 | 328 | 342 | | | | | 330 | 8 |

**Table 1:** Water depth below current lake level for each identified delta. Profiles issyk0049, ik30 and ik28 are located on the western and all other profiles are located on the eastern delta. Two-way traveltime was converted to water depth using a sound velocity of 1500 m s$^{-1}$. mean = mean value, std dev = standard deviation calculated for each delta lobe including measurements from the eastern and western areas where available.



**Fig. 1: Geographical settings** of Lake Issyk-Kul. Tracklines of all profiles used in this study are shown in green (data from 1997) and orange (2001). The axes of two anticline structures visible in the southeastern part of the lake are marked in red. Grey hatched areas mark the approximate position of the Issyk-Kul Broken Foreland (after Macaulay et al., 2014).

**Fig. 2: N-S profile ik01**. The asymmetric nature of the lake basin is clearly visible with considerably higher subsidence rates towards South.

**Fig. 3: Facies types** of Lake Issyk-Kul sediments.

**Fig. 4: Seismic profile issyk019**. Upper panel: Seismic data. Lower panel: Linedrawing and interpretation of all sequences identified in the profiles. Trackline of profile issyk019 is marked in Fig. 1.

**Fig. 5: Sequences 6 and 7 on seismic profile issyk019**. Upper panel: Sequence 6. Lower panel: Sequence 7. Seismic data overlaying the respective sequences is removed.

**Fig. 6: Sequences 3 / 4 and 5 on seismic profile issyk019**. Upper panel: Sequence 3 and 4. Boundary between these two layers cannot be identified; it is even possible that sequence 3 was eroded and only sequence 4 is visible. Lower panel: Sequence 5. The missing (eroded) parts of deltas 5.1 and 5.2 are tentatively completed in light colors for better visualization. Seismic data overlaying the respective sequences is removed.

**Fig. 7: Sequences 1 and 2 on seismic profile issyk019**. Upper panel: Sequence 1 with distinct deltas 1.1 and 1.6 and the transgressional phase. Deltas 1.2 to 1.5 are only visible in parallel profiles, but their relative location is marked. Note that the lower boundary of Sequence 1 is highly erosive with deeply incised channels. Lower panel: Sequence 2. The missing (eroded) parts of deltas 2.1 and 2.2 are tentatively completed in light colors for better visualization. Seismic data overlaying the respective sequences is removed.

**Fig. 8: Lake-level curve** of Lake Issyk-Kul. Red line: mean value of delta depths; gray shading: standard deviation of delta depths. Numbers correspond to delta numbers in the text and in table 1. Boundaries in between the different sequences are at least partially erosive.







**Figure 1**





**Figure 2**



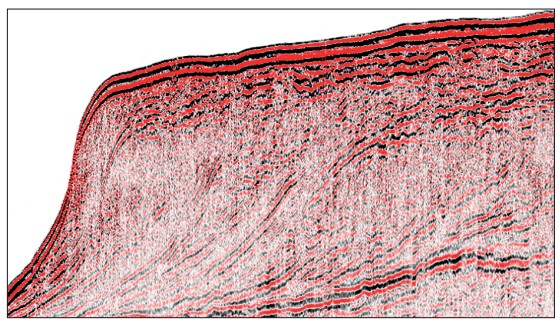

**a. Facies I:** Delta with topset and foreset

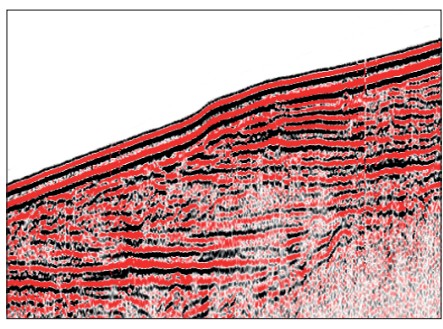

**c. Facies III:** Transgressional deltaic sediments

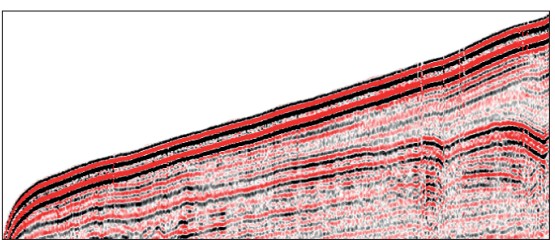

**b. Facies II:** Prodelta sediments

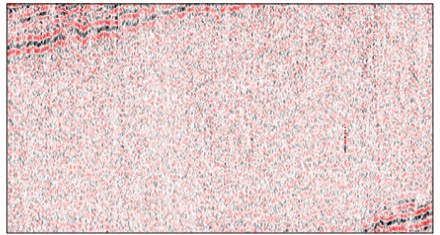

**d. Facies IV:** Acoustically transparent

**Figure 3**



**Figure 4**







**Figure 5**





**Figure 6**

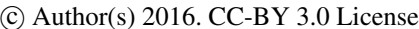





**Figure 7**





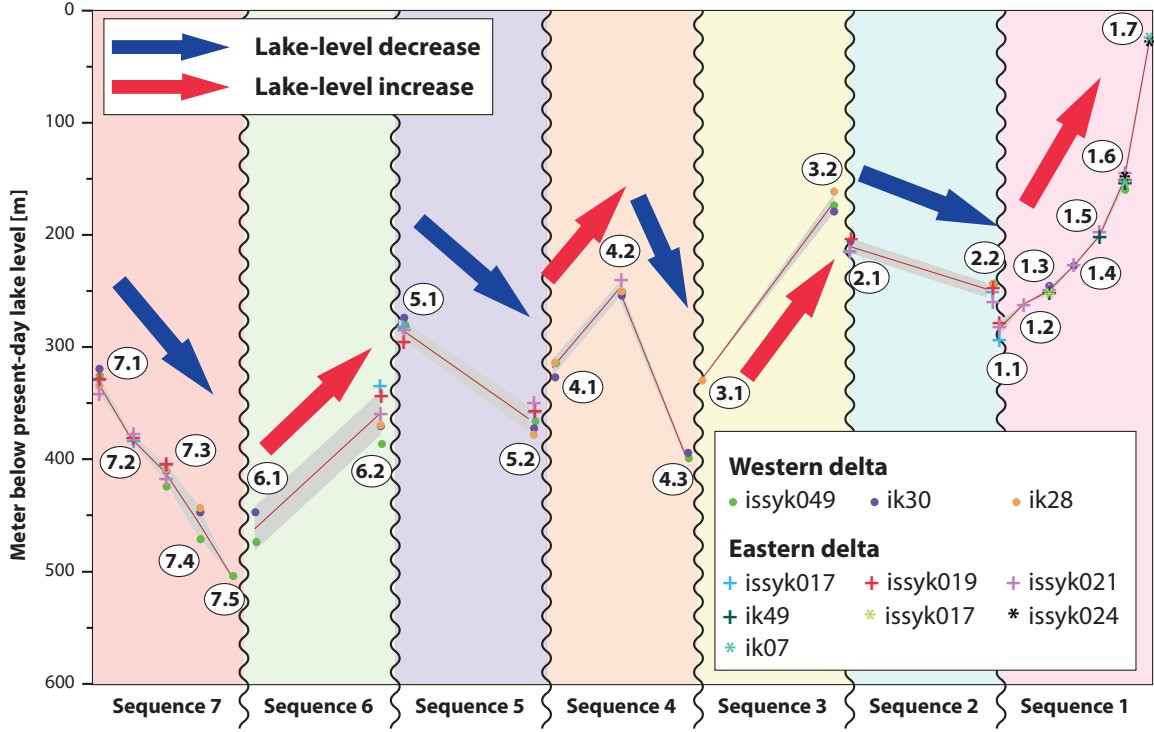

**Figure 8**