# Peer review of "High-amplitude lake-level changes in tectonically active Lake Issyk-Kul (Kyrgyzstan) revealed by high-resolution seismic reflection data"

_Climate of the Past, 2016_

## Referee Comment (RC1) · Anonymous Referee #1 · 19 Feb 2016

The manuscript 'An extended history of high-amplitude lake-level changes in tectonically active Lake Issyk-Kul (Kyrgyzstan), as revealed by high-resolution seismic reflection data' by Gebhardt et al presents a detailed and interesting analysis of lake level fluctuations of Lake Issyk-Kul and links the fluctuations to past changes in the atmospheric circulation pattern. This is an interesting aspect because long climate archives from the investigated area are sparse. Unfortunately, no age information are available, which does not allow linking the circulation patterns to specific periods. However, the conclusion that a cyclic pattern caused by changes in the atmospheric circulation pattern exists, is significant and should be published. Hence, I strongly recommend

publication of this manuscript. However, several modifications are needed prior to final publication. My main concern is the poor data description and documentation. Only one profile crossing deltaic features is shown. The reconstruction of the lake level fluctuations is difficult to follow, as many deltaic sequences are not shown on this profile. I am aware that not all deltas can be shown but it would be good to present some more data, which would at least proof that structures are similar at the western and eastern end of the lake. This is mentioned in the manuscript but not supported by any presented data. These profiles should be described first, which may then act as basis for the interpretation. I am not a native English speaker; hence, I have not made any language corrections.

My main general points of critics are: 1) Show more data and give a better general description of the data. The detailed description of the stratigraphic units as presented now is not really a description. This is more a stratigraphic interpretation, which is not based on a proper description. Fig. 4 can be used for a general description but this figure is even not referenced in the text at all. I suggest to show one profile from the eastern and western parts, each. These profiles should be described first. Explain how you define the stratigraphic sequences in general. Point to the similarities (and diffeences) between the eastern and western area. Mark all the deltaic sequences.

2) Carefully check the usage of terminology for the seismic stratigraphic description/interpretation. E.g., you write that you have erosion at the upper and lower boundary of a unit. Per definition, an erosional truncation is termination of reflectors against an upper boundary caused by erosion. It may well be that both boundaries show erosional features, but then you need to carefully describe, that you have erosional truncation of the unit below the sequence boundary and downlap/baselap/onlap/conformity above the boundary. When describing unconformities, always describe termination above and below the unconformity.

3) You define the topset-foreset roll-over point as a proxy for the lake level at the time of its formation. This is a valid approach. Based on the distribution of the clinoforms,

you can conclude that you had rapid lake –level changes. However, I do not agree that all clinoforms/deltas have been deposited at times of lake-level stillstands. Some of the clinoforms look like forced regression-system tracts, indicating a falling lake level. Others may show some aggradational patterns indicating a slow lake level rise. I agree that the clinoforms indicate relatively constant lake levels or only small changes compared to the rapid changes documented by the different locations of the delataic sequences.

4) Distribution of delta sequences. In order to reconstruct the lake level fluctuations, you use many delta-sequences not shown on your seismic example. You state that most delta sequences have been identified on both sides of the lake but this is not documented. You even do not mark all delta sequences identified on the presented line (Figs 4-7, you list much more in Table 1). Mark them. Why do you have such an incomplete record of deltas on single lines? I assume that this is caused by changing points of sediment input to the lake (as partly discussed in the manuscript) but you should mention this somewhere (distribution of deltas and what causes lateral shifts of deltas).

5) Good overview map is missing showing the general location of the lake and its tectonic setting (I doubt that most people would be able to place the lake on a world map). Many locations are given in the text, which are not shown on any figure. Link between text and figures should be improved.

6) It would be nice to include a small outlook in the conclusions. You clearly state that it would be important to date the delta features in order to establish a solid link between climate and deposited sequences. Come back to this point in the conclusion.

Below you find more specific comments for each chapter and the figures. Good luck!

Below you find more specific comments for each chapter and the figures. Good luck!

Specific comments:

Abstract:

P1, Line 18: Change 'identify' to 'reconstruct'

P1, Line 19: See general comment concerning lake level still stands. Lake level was relatively constant compared to quick fluctuations in other periods.

P1, Line 21: Delete 'during the past'

Introduction P2, Line 8: Summarize the previous statements. Something like. The examples demonstrate that lake level fluctuations may document climate change (regional and/or global), changes in basins morphology and barriers, as well as tectonic and volcanic forcing.

P2, Line 9: Refer to figure showing the general location of Lake Issyk-Kul in a broad context. Such a figure is missing. This figure should also include all regional features/locations you mention in the text.

P2, Line 14, 15. Split last sentence to two sentences.

Study area

P2, Line 18-22: Make sure that all locations are shown on a good overview map.

P3, Line 2: Refer to your figures. The link between text and figures should be improved.

P3, Lines 2 – 5: Split sentence to two sentences.

P3, Line 8: Explain how the shelf is separated from the slope.

P3, Line 20: Was the lake ice-covered during the glacials? Did glaciers cover the shelf? Any information?

P5, Line 3: Show Tien Shan Mountains on overview map. Also true for other locations and not mentioned again in this review. Check carefully.

P5, Line 14, 15: Split to two sentences.

Data acquisition and processing

P6, Line 10-14: This statement should be supported by a figure (in the result section).

Results and Interpretation

P6, Line 22: Penetration of 375 m is not documented on any figure.

P6, Line 26: The MTF should be marked on an overview figure.

P7, Line 2: Correct 'becaoem'

P7, Line 3: 'The anticline' has not been introduced before. Some information is needed.

P7, Line 6: I agree that deformation is still active but I cannot see that the uppermost layers still display a slight dip angle.

P7, Line 9: For which period are the sedimentation rates valid? Are they only valid for the Holocene as they are based on short cores? Can you really use them as mean rate for calculating age? You partly comment on this further down but I would expect significant variations of sedimentation rate between humid and arid climatic phases.

P7, Line 19/20: Are these anticlines visible on your data. Not clear. Make clear what results are based on your data.

P8; Line 23: I cannot see the onlap on the figure.

P9, Line 19: Change 'forming' to 'formation'

P9, Line 22 and following: How is the upper boundary of this unit defined?

P10, Line 1-5: Give reference to figures. On Fig 5b, no delta is marked despite the fact that Tab. 1 suggests that delataic sequences 7.1., 7.2 and 7.3 should be visible. I may see one delta but it remains unclear where you interpret the other deltas. Mark all interpreted deltas very clearly on the figure.

P10, Line 8: A sequence may have erosional truncation as upper boundary but not as lower boundary. Hence, the statement that the sequence exhibits erosive upper and lower boundaries is not precise. As for sequence 7, no deltas are marked on Fig. 5 for

sequence 6.

P10, Line 12 and following: When describing unconformities, always describe termination above and below the unconformity. The upper boundary of unit 6 shows erosional truncation. I cannot judge the characteristics of the lower boundary of sequence 5. Check very carefully for the description of all sequences. I will not comment on this for the other units. For delta 5.2. I do not see details but it seems to be a forced regression and not a real stillstand.

P10, Line 21 and following: How do you explain the pronounced step in the morphology of the upper boundary of Sequence 3/4?

P11, Line 8: Sequence 3 may fill erosional features but the lower boundary is not erosive. It is the upper boundary of the underlying sequence. Check also for other sequences.

P11, Line 26: This is a correct description (It lies above an erosional unconformity and sediments fill the channels).

P12, Line 3: I assume it should be Sequence 1 (and not 2)

Discussion P13, L3: See general comment concerning lake-level stillstands.

P13, Line 28, Boom gorge has not been introduced before. Refer to Fig. 1, where it is shown.

P15, .Line 10: See previous comment about the anticlines (P7, Line3).

P15, Line 25: What do you mean with 'May have influenced' Again, no detailed information about the anticlines is given in the manuscript. The anticlines are not critical for the manuscript but you draw conclusions based on the anticlines without a real presentation of these anticlines.

P16, L1-5: see general comments. Should be illustrated in a figure.

P16, Line 22 – 28: This has already been partly discussed in the previous section but it also partly contradicts the previous section, where it is stated that subaerially exposed terraces may indicate lake levels 100 m higher than present. Clarify.

P18, Line 3, 4: Is there a reason that you are nor listing rainfall/direct precipitation?

Page 18, Line 15: See general comments about lake-level stillstands.

Conclusions

P19, Line 16: Change to 'each stratigraphic section contains at least 2 . . .'

Figures:

Fig. 1: An overview map showing the general location of the lake and regional features is missing. Colour code would be useful. The profiles shown in the manuscript should be marked much clearer (direct reference to the Figure).

Fig. 2: Depth below lake floor scale is a bit confusing. How have you set the zero point? I would recommend changing the scale to depth beneath present lake level.

Fig. 3: OK. If you show a profile from the western shelf, you should mark some of the prominent deltas identified on both profiles.

Figs. 4-7: See comments above. You need to mark all deltas identified on this figure. Much more delta features than marked on the figure are listed in Table 1 for this profile. There is no reference to Fig. 4 in the text.

Fig. 8: OK

---

## Referee Comment (RC2) · Anonymous Referee #2 · 20 Feb 2016

General comments (comment refer to page number/line number where indicated)

This is a well presented and scientifically significant study that is very appropriate to be published in the 'Climate of the Past' journal. The interpretations are sound, well-based on data and provide new insights into a highly dynamic paleoclimate regime in Central Asia. Eventhough the data do unfortunately not allow a dating of the presented wet-dry climate cycles ('a reason to drill the lake'), the presentation of these patterns nevertheless provide novel data that are absolutely worth to be reported.

My main concern refers to limited amount of data shown as figures: The figures (beautifully crafted by the way, a pleasure to look at) focus basically on a single seismic line. The line is spectacular indeed, the seismic stratigraphic interpretation sound and somehow textbook-style, but that same seismic line is shown on 4 full pages with different levels of annotations, way to much. There is no need to show every infilling step of each seismic sequence, the only added value comes in Figs. 6b and 7b, where eroded sequences are reconstructed, but that can also go in a smaller extra figure. What is needed much more are more shown examples. I am curious how representative this singled-out seismic line really is. In fact, many of the discussed delta lobes are not presented but provide crucial elements of the lake-level reconstruction. As reader, I need to see at least 2-3 more examples of seismic lines from other areas of the lake (for instance also the Western delta area), i.e. more of the sequence stratigraphic architecture. This can be done at 'no cost', as Figs. 4-7 can be reduced to one full page, there is plenty of space available. Having said this, I also would appreciate with new figures or maybe also in map view what is really meant with the concept of 'delta lobes' and how they are distributed on both sides of the elongated lake. These lobes, and their vertical and lateral stacking pattern is the key to reconstruct the details of the lake-level curve, so these data are crucial but yet not presented.

I am intrigued by the fact that all sequences and their boundaries on the shown seismic line display a gentle basinward dip. Is this a pattern on all seismic sections, also in the West? Or is this formed by a general forced regressional pattern with falling lake level upon delta progradation? But why is there never an still stand (horizontal progradation) or even an aggradation of a delta sequence upon a gently rising lake level? Is this a function of tectonic subsidence or tilting?

I am also wondering why sequence 5 is not subdivided in two main sequences (currently called subsequences 5.1 and 5.2), as they are separated by a very clear unconformity. What defines the hierarchy of the sequences? On contrast, I am not fully convinced that sequences 2.1 and 2.2 represent clearly two pulses or whether they form a transitional package without major unconformity in-between them. Both of these issues

are hard to track, as one shown seismic line alone from the shelf is not sufficient.

Discussion on p.14 about deltas 1.1-1.6 is hard to follow. I cannot judge on he basis of the limited shown data whether 1.1-1.6 is indeed in chronologic order or whether lake level plays as a 'jojo' reshuffling the lobes in maybe a different order? Moreover, the arguments presented for an uplifted nature of the subaerially exposed terraces are a bit weak, I am somehow not convinced in this matter.

Further comments

The English language can be improved in some of the sections....maybe have an English native speaker go through it.

Shorten title by deleting 'An extended history of', just start with 'High-amplitude lake-level fluctuations of....'

1/21: delete one of the two 'past'

1/22: ....from the Mongolian steppe blocking the mid-latitude Westerly's.

2/6: ... AND thermal expansion...

2/6: HenCe (spell checker!!)

2/6: no comma after curve

2/10: Three 'large' on one line, too much!

2/25: The quoted publication (Anselmetti et al., 2006) initially stated indeed glacial/interglacial cycles, which after drilling turned out to be stadial/interstadials, I would change to:....were correlated to wet-dry paleoclimate patterns with lowstands during the stadials and highstands during the interstadial periods (Hodell et al., 2008, Quaternary Science Reviews 27, 1152-1165)

3/2: Lake-level changes as large as 170 m have on one hand been attributed to....

3/5: Awkward short sentences, change to: The impact crater of Lake Bosumtwi

(Ghana) is....

3/8: ...is purely driven by the evaporation/precipitation ratio.....

3/9: Lake Issyk-Kul, subject of this study, is....

3/21: Figure 1 shows these mountains exactly reversed (N vs. S). Which one is correct?

4/4:' ...and by steep...' poor English, unclear what is meant, reword or make 2 sentences

4/7: can this 110 m depth transition be marked on Fig. 1

4/21: thRough

4/22: Surface-water temperature...

4/26: This is a hydrologically 'bold' statement....any references?

5/16: 2004 is not 'recently'

6/14: avoid one-sentence paragraphs.

Fig. 2: I note a somehow prominent change in basinal sediment geometry (draping vs. filling) at ∼1.1 s twt in the middle of the profile, in particular when correlating to the right side of the figure...is this worth to be discussed?

Make sure final Fig. 2 has sufficient resolution, I have problems seeing for instance the mass-transport deposit.

7/2: became (Spell checker!!)

7/4: Which anticline? Has no been mentioned before

7/4:...dip angle OF the strata....

7/12: two 'however' within 4 words:-(

7/12: But on the figure I only see ca. 200 m of sediments.....the authors report partly 370 m sediment thickness but no evidence is shown.

7/20-25: The longer anticline still is visible on lake floor, correct? It looks like a dipping anticline (towards SW), is that worth to be mentioned? Are the two anticlines aligned in an 'en echelon' pattern? Are these anticlines really tectonic in nature of simply a draping remnant of an underlying basement high?

8/7 and ff.: Use throughout the manuscript 'reflections' instead of 'reflectors'. On seismic data, you only see reflections. Reflectors (=impedance change in the sediment record) cannot have amplitudes.

Seismic facies 3, here the term 'retrogradation' may also be used, or a 'backstepping' delta.

13/6: ...riseS...

13/9:..mainly in the shallower parts of the lake

14/7: this 'some' here and in numerous other places in the text is not elegant: use 'ca.' or even better a '∼'.

14/25: I don't agree with the mentioning of the outflow here: The balance is made by precipitation/inflow and evaporation only (maybe subsurface outflows). The outflow is a result of positive hydrologic balance, thus the difference in the balance, then the lake level is geomorphologically fixed and the system open. If the balance is negative, then the outflow will be zero and the basin closed.

15/10: No one-sentence paragraph

15/20ff: Why don't the authors call the system a half-graben? with the main border fault in the south? It has all indications, correct?

Time constrains on 15/20: suddenly the term 1 Ma pops up? What is the origin? based upon? No age data at all has been presented before!

Discussion on highstands is necessary but what about tectonics as regulator of outflow level? 16/25 ff: It also could suggest that the outflowing area subsided relative to the lake, lowering the topographic outflow point.

The last figure and the general lake-level reconstruction based on 'shallow' sedimentary sequences is highly reminiscent to another study in a Patagonian hydrologically closed lake where the first- and one co-author were also co-authoring: I would also quote this study, as some of the concepts match very nicely (Anselmetti et al., 2009, Sedimentology 56, 873-892)

One should remove the thick red and blue arrows on last figure and make lake-level lines thicker, that will be much better to visualize these impressive lake-level variations,

---

## Referee Comment (RC3) · Anonymous Referee #3 · 22 Feb 2016

Dear author, I have completed my review of "An extended history of high-amplitude lake-level changes in tectonically active Lake Issyk-Kul (Kyrgyzstan), as revealed by high-resolution seismic reflection data" by Gebhardt et al. The manuscript presents high-resolution sparker profiles from Lake Issyk-Kul and has the potential for being a broad and useful study. I personally enjoyed reading it and highly recommend for its publication. Though, it will need to undergo major revisions before it is acceptable for publication. In my review, I outline both major critiques and minor points in the lists below.

[Figure]

Major Points:

-I strongly suggest the authors to present more seismic profiles from different parts of the lake; this is indeed lacking in the current manuscript. In particular, profiles showing deltas from the western margin would be great in order to compare their internal/external structure with the ones from eastern part of the lake.

-Significant lacking of citations in the results part. The authors, most of the times, do not cite or refer figures in the text. Sometimes, the figures are not large enough to see points menntioned in the text, for instance erosional boundaries, delta lobes. Hence, as a reader it is rather difficult to judge the interpretation.

-Would it be possible to correlate stratigraphic boundaries towards the deeper parts of the lake? I can see that deep lake sediments are characterized by alternating high- and low-amplitude seismic reflections which most likely reflect transgression and regression periods.

-I am also missing isopach or isochron maps of seismic units in order to understand their thickness variations and thus the source regions through lake evolution. If this is not applicable or doable, it is better to mention the average thickness of individual units and possible source regions in the text.

- I suggest the authors to make a new basemap and draw lakeward boundary of the deltas (color-coded) in order to see their lateral extent along the western and eastern shelves. The distribution of sublacustrine channels can also be superimposed on this map.

-The authors presents and discuss structural setting of the lake, however I do not see any structural map showing faults, anticlines, or synclines throughout the lake as well as its surroundings. I see several seismic profiles crossing the anticline structures on the base map but neither of them is shown. It is worth to discuss the relative timing of these structures based on thickness variations of overlying/underlying sediments. Also

a normal fault in the southern part of Profile ik01 (Fig. 2) should be shown.

-I suggest the authors make schematic diagrams (with scale) from East to West showing the formation of deltas throughout the lake formation. The former lake levels should be indicated. This would definitely improve the quality of the manuscript.

Line points:

-Page 6, Line 24. "..presence of a series of faults.." It would be better to show these faults on a map.

-Page 6, Line 25-26. Please locate the "Main Terskey Fault (MTF)" on a map.

-Page 7, Line 2. Change "becaoem" to "became " and "Miocene" to "Miocene"

-Page 7, Line 12. Modify so that it reads, "..However, it is quite likely that . . ."

-Page 7, Line 18-19. "In the southeastern part of the lake, the strata are not inclined as would be expected in this asymmetric basin". Please refer to figure or show a seismic section. I can see that there are various seismic profiles traversing these anticline structures.

Page 7, Line 23-24. "Both anticlines are progressively buried by younger sediments, and the southern one is meanwhile completely leveled by sediments." Please show a seismic section as I cannot confirm whether they are buried or leveled by sediments.

Page 8, Section 4.2 Facies Types. I suggest changing "Facies types" as "Seismic Facies" and the "Facies I" as "Seismic Facies 1 (SF1)". It is easier for descriptions. It is also better to formulate as "SF1 is characterized by. . . " than "this facies type is characterized. . ."

Page 8, Line 11. Clinoforms should be better illustrated; topsets-foresets transitions (if they exist) are not noticeable on the presented seismic profiles. I propose to the authors to add a figure as an example of interpreted delta (for instance, immediately below Fig. 3a; (3b, interpreted section of 3a) in which reflections of topset, foreset and

bottomset are pointed.

Page 9, Lines 1-2. "Facies IV may be interpreted as former delta sediments that have been affected by post-depositional processes (e.g. sediment remobilization, slumping, liquefaction) that caused them to loose their internal structure". They are indeed former delta deposits but I am not sure such reflection configuration was caused by slumping etc. Looking at reflections within Sequences 3/4 in Fig. 4, as a whole package, I do not think it has something to do with slump deposits. Would it be possible that such reflections were due to coarse-grained sediments resulted from rapid loading of rivers?

Page 9, Line 18-19. "The topset-foreset roll-over point is considered as a proxy for the lake level at the time of its forming". Please give a reference.

Page 9, Line 23. Modify so that it reads, "Sequence 7 (S7) is the..." In the following parts you can shorten its name as "S7" instead of "Sequence 7".

Page 9, Line 16. Please delete "lacustrine"

Page 10, Line 2. "..Some of these occur only in the western delta area (7.5, 7.4)" Please refer or show a seismic section.

Page 10, Line 7. "Sequence 6 is clearly visible both on the western and eastern delta areas." Please refer to Fig.

Page 10 , Line 8. "... rather thin..." How much?

Page 10, Line 9. ".... delta lobes could be identified at 461 (no. 6.1) and 361 m bll (6.2)." Could you please label these delta lobes in the seismic sections?

Page 10, Line 13. "Sequence 5 is overlaying sequence 6 with an erosional boundary in between (Fig. xx??).

Page 10, Line 14-15. "The bathymetrically higher delta 5.1 exhibits extensive erosion (Fig. 6b)". I am looking at this figure and it is almost impossible to see the erosional surface. I suggest the authors to show close-up sections to show these features.

Page 10, Line 21. Modify so that it reads, "Sequence 4 overlies . . ."

Page 10, Line 24-25. Modify so that it reads, "In Sequence 4, the delta lobes are characterized by predominantly SF4, but. . ."

Page 10, Line 26. Change "well-layered" into "well-stratified". Please also make the colors of sequences more transparent so that the internal reflections can be seen clearly.

Page 11, Lines 1-2. "Three delta lobes were identified: the oldest (4.1) at ca. 319 m bll, followed by a delta (4.2) at approximately 250 m bll and a third, younger (4.3), at 397 m bll." Where are they in the seismic section? Please mark the locations of these deltas.

Page 11, Line 6. " Sequence 3 could only be clearly identified in the western delta areas;. . ." Please show a seismic profile from the western area which clearly depicts S3.

Page 11, Lines 8-9. "In the western delta complex, Sequence 3 is characterized by a lower boundary that was partially erosive into the underlying sediments but grades into a correlative conformity in other places." I cannot judge this interpretation as I do not see any figure showing this relationship.

Page 13, Lines 4-5. Instead of using lake level decrease and increase, how about using regression and transgression?

Page 14, Line 3. Change "Subaquatic channels" into " Sublacustrine channels"

Page 15, Line 18. ". . .subsidence seems to have been relatively constant through time." Can you quantify the fault activity by looking at thickness variations towards it?

Figure Captions Overall, the figure captions should be improved.

Figures

Fig. 1. Please add an inset map showing large areas of the regions. With the current

map, I cannot say where the Lake Issyk-Kul is located. The depth color bar is missing as well. What about the bathymetry of the lake reconstructed from seismic reflection profiles?

Fig. 2. I suggest including vertical exaggeration for all seismic profiles. Locate the fault on the southern end of the profile. Can you please enlarge the MTDs?

Fig. 3. Please add vertical and horizontal scales.

Fig. 4. It would also be better to give names for the sequence boundaries, such as Sequence boundary 1 (SB1) to SB7. But it is your choice.

Fig. 5. Please switch the Figure 5a and 5b. It should be displayed in an order and should start from Sequence 7. Please do this for the following figures.

Regards

———————————————————

---

## Author Comment (AC1) · 3 May 2016

Dear reviewers,

many thanks for your highly valuable comments! I notice that all three reviewers share the main concerns and suggestions for revision. I therefore sum these up in this first part of my answer. In the second part I will respond to individual comments by the reviewers.

First part:

[Figure]

Suggestion 1: More seismic profiles All three reviewers suggest to show more seismic profiles, at least one from the eastern and one from the western delta. In turn, some of the current figures can be summed up in one figure. I agree to this point and will add more seismic profiles. In fact, parallel profiles from the eastern delta are very similar to the one shown, but this is valuable information for the reader, too. Profiles from the western delta are comparable but slightly less textbook-style as those from the eastern delta. I will also add a chapter with a description of the seismic profiles as suggested by Reviewer #1. Furthermore, I will also show more detailed views from the seismic profiles, e.g. from the stratigraphic boundaries to show the truncation of the reflectors, and name them correctly as downlaps/onlaps etc. as was suggested by Reviewer #1. I will check for a sequence boundary (or its absence) between 5.1 and 5.2, and for the transition between 2.1 and 2.2 as suggested by Reviewer #2, and change text/interpretation if appropriate. I will furthermore check the clinoforms/deltas for indications of a forced regression system tract, and change the lake-level curve where necessary (Reviewer #1). Yes, all shown sequences and their boundaries do show a gentle basinward dip (Reviewer #2). I will go through all other profiles again and check if this is truly consistent throughout all profiles. I shall discuss this in the revised version of the manuscript. In this paper, however, I will concentrate on the (climatically-induced) lake-level variations and only focus on tectonic/structural features where it is absolutely necessary to (a) understand the profiles or (b) distinguish between tectonically-driven and climatically-induced lake-level variations (Reviewer #3). A second paper is currently prepared by a student that focuses on the tectonic nature, the location of faults, their relative timing, etc. in this lake – this is a full story on itself. I don't think that the full spectrum of tectonic features visible in the lake basin is necessary in this current manuscript to understand the paleoclimate history of the lake, and hence do not want to jeopardize the student's paper.

Suggestion 2: Map with the location of the delta lobes etc./with isopachs/isochrons I was thinking of such a map already for the current version of the manuscript, but was hesitant to compile one. My major concern is that while we can identify delta sequences/delta lobes in many profiles, they are absent in others due to two reasons: Either because they were never deposited, or because they were eroded. A map on the currently visible features will however not differentiate between those that were not deposited and those that were deposited but later eroded. A map will thus underestimate the true extent. This is also true for a map of thickness of the sequences. I think this is best visualized in current Figs. 5b and 6b: Using the current thickness of the sequences would definitely underestimate the true and initial thickness of these sequences. I will nevertheless add a map to the revised version of the manuscript to show the general extent of the delta sequences. I agree that this will enhance the understanding of the reader about the spatial extent of these features significantly. At the same time, I will also add a paragraph where I discuss my concerns on this issue.

Additionally, I will definitely add an overview map showing the geographical location of the lake, geographical names used in the manuscript, and add scales, color bars, etc. to the figures where appropriate/missing.

Second part:

Reviewer #1: I will add an outlook paragraph on the importance to date the delta sequences, and hence to date the paleoclimate record. The lake is still one of the drilling targets of ICDP which I agree should be mentioned in the manuscript. I will go through all smaller comments to specific lines in the manuscript, thanks for this very detailed work!

Reviewer #2: I will change the discussion on deltas 1.1 to 1.6 and add a detailed view of this part of the seismic line. On what concerns the subaerially exposed terraces: Unfortunately we do not have own data but can only use what is published so far. I will try to clarify this better in the text and re-think my own interpretation and concerns on this matter. Thanks for all the smaller comments that I will include in the revised version of this manuscript. I did not know that the Anselmetti et al. 2006 was updated by Hodell et al. in 2008 – I will definitely change this. On what concerns the terminology

"halfgraben": My co-author Ed Sobel is carrying out tectonic work in this region. He did not agree with the term halfgraben for the lake basin that I used in a former version of this manuscript.

Reviewer #3: I will more carefully check the referencing of the figures and I will add detailed views on crucial parts of the profiles. Correlation between the delta sequences and the layering visible on the lake floor is unfortunately impossible due to the steep flanks where layers are too thin to be correctly traced. To our current knowledge the alternation between low and high amplitudes in the central part of the lake is due to frequent turbidites (which may or may not be reflecting transgressional and regressional periods). Your suggestion of adding a schematic diagram from East to West showing the formation of deltas throughout the lake formation, and including the former lake levels is highly appreciated. I think this would really improve the understanding of the lake's evolution. Thanks! Also your suggestion to abbreviate e.g. the seismic facies types to SF and Sequence to S and to add names for the Sequence boundaries will be included in the revised version of the manuscript. All smaller comments will be checked and used to improve the manuscript.

---

## Editor Comment (EC1) · V. Rath (Editor) · 24 May 2016

Dear Authors, this review has been positively reviewed by three referees, however, two of them require major revisions. I think the necessary moderate changes - mainly addition of material - can easily be included in the manuscript for CP publication. I therefore ask you to submit a revised manuscript. Regards, Volker Rath

---

## Author Response (AR1)

The manuscript 'An extended history of high-amplitude lake-level changes in tectonically active Lake Issyk-Kul (Kyrgyzstan), as revealed by high-resolution seismic reflection data' by Gebhardt et al presents a detailed and interesting analysis of lake level fluctuations of Lake Issyk-Kul and links the fluctuations to past changes in the atmospheric circulation pattern. This is an interesting aspect because long climate archives from the investigated area are sparse. Unfortunately, no age information are available, which does not allow linking the circulation patterns to specific periods. However, the conclusion that a cyclic pattern caused by changes in the atmospheric circulation pattern exists, is significant and should be published. Hence, I strongly recommend

publication of this manuscript. However, several modifications are needed prior to final publication. My main concern is the poor data description and documentation. Only one profile crossing deltaic features is shown. The reconstruction of the lake level fluctuations is difficult to follow, as many deltaic sequences are not shown on this profile. I am aware that not all deltas can be shown but it would be good to present some more data, which would at least proof that structures are similar at the western and eastern end of the lake. This is mentioned in the manuscript but not supported by any presented data. These profiles should be described first, which may then act as basis for the interpretation. I am not a native English speaker; hence, I have not made any language corrections.

My main general points of critics are: 1) Show more data and give a better general description of the data. The detailed description of the stratigraphic units as presented now is not really a description. This is more a stratigraphic interpretation, which is not based on a proper description. Fig. 4 can be used for a general description but this figure is even not referenced in the text at all. I suggest to show one profile from the eastern and western parts, each. These profiles should be described first. Explain how you define the stratigraphic sequences in general. Point to the similarities (and diffeences) between the eastern and western area. Mark all the deltaic sequences.

2) Carefully check the usage of terminology for the seismic stratigraphic description/interpretation. E.g., you write that you have erosion at the upper and lower boundary of a unit. Per definition, an erosional truncation is termination of reflectors against an upper boundary caused by erosion. It may well be that both boundaries show erosional features, but then you need to carefully describe, that you have erosional truncation of the unit below the sequence boundary and downlap/baselap/onlap/conformity above the boundary. When describing unconformities, always describe termination above and below the unconformity.

3) You define the topset-foreset roll-over point as a proxy for the lake level at the time of its formation. This is a valid approach. Based on the distribution of the clinoforms,

[Figure]

you can conclude that you had rapid lake –level changes. However, I do not agree that all clinoforms/deltas have been deposited at times of lake-level stillstands. Some of the clinoforms look like forced regression-system tracts, indicating a falling lake level. Others may show some aggradational patterns indicating a slow lake level rise. I agree that the clinoforms indicate relatively constant lake levels or only small changes compared to the rapid changes documented by the different locations of the delataic sequences.

4) Distribution of delta sequences. In order to reconstruct the lake level fluctuations, you use many delta-sequences not shown on your seismic example. You state that most delta sequences have been identified on both sides of the lake but this is not documented. You even do not mark all delta sequences identified on the presented line (Figs 4-7, you list much more in Table 1). Mark them. Why do you have such an incomplete record of deltas on single lines? I assume that this is caused by changing points of sediment input to the lake (as partly discussed in the manuscript) but you should mention this somewhere (distribution of deltas and what causes lateral shifts of deltas).

5) Good overview map is missing showing the general location of the lake and its tectonic setting (I doubt that most people would be able to place the lake on a world map). Many locations are given in the text, which are not shown on any figure. Link between text and figures should be improved.

6) It would be nice to include a small outlook in the conclusions. You clearly state that it would be important to date the delta features in order to establish a solid link between climate and deposited sequences. Come back to this point in the conclusion.

Below you find more specific comments for each chapter and the figures. Good luck!

Below you find more specific comments for each chapter and the figures. Good luck!

Specific comments:

Abstract:

[Figure]

P1, Line 18: Change 'identify' to 'reconstruct'

P1, Line 19: See general comment concerning lake level still stands. Lake level was relatively constant compared to quick fluctuations in other periods.

P1, Line 21: Delete 'during the past'

Introduction P2, Line 8: Summarize the previous statements. Something like. The examples demonstrate that lake level fluctuations may document climate change (regional and/or global), changes in basins morphology and barriers, as well as tectonic and volcanic forcing.

P2, Line 9: Refer to figure showing the general location of Lake Issyk-Kul in a broad context. Such a figure is missing. This figure should also include all regional features/locations you mention in the text.

P2, Line 14, 15. Split last sentence to two sentences.

Study area

P2, Line 18-22: Make sure that all locations are shown on a good overview map.

P3, Line 2: Refer to your figures. The link between text and figures should be improved.

P3, Lines 2 – 5: Split sentence to two sentences.

P3, Line 8: Explain how the shelf is separated from the slope.

P3, Line 20: Was the lake ice-covered during the glacials? Did glaciers cover the shelf? Any information?

P5, Line 3: Show Tien Shan Mountains on overview map. Also true for other locations and not mentioned again in this review. Check carefully.

P5, Line 14, 15: Split to two sentences.

Data acquisition and processing

[Figure]

P6, Line 10-14: This statement should be supported by a figure (in the result section).

Results and Interpretation

P6, Line 22: Penetration of 375 m is not documented on any figure.

P6, Line 26: The MTF should be marked on an overview figure.

P7, Line 2: Correct 'becaoem'

P7, Line 3: 'The anticline' has not been introduced before. Some information is needed.

P7, Line 6: I agree that deformation is still active but I cannot see that the uppermost layers still display a slight dip angle.

P7, Line 9: For which period are the sedimentation rates valid? Are they only valid for the Holocene as they are based on short cores? Can you really use them as mean rate for calculating age? You partly comment on this further down but I would expect significant variations of sedimentation rate between humid and arid climatic phases.

P7, Line 19/20: Are these anticlines visible on your data. Not clear. Make clear what results are based on your data.

P8; Line 23: I cannot see the onlap on the figure.

P9, Line 19: Change 'forming' to 'formation'

P9, Line 22 and following: How is the upper boundary of this unit defined?

P10, Line 1-5: Give reference to figures. On Fig 5b, no delta is marked despite the fact that Tab. 1 suggests that delataic sequences 7.1., 7.2 and 7.3 should be visible. I may see one delta but it remains unclear where you interpret the other deltas. Mark all interpreted deltas very clearly on the figure.

P10, Line 8: A sequence may have erosional truncation as upper boundary but not as lower boundary. Hence, the statement that the sequence exhibits erosive upper and lower boundaries is not precise. As for sequence 7, no deltas are marked on Fig. 5 for

sequence 6.

P10, Line 12 and following: When describing unconformities, always describe termination above and below the unconformity. The upper boundary of unit 6 shows erosional truncation. I cannot judge the characteristics of the lower boundary of sequence 5. Check very carefully for the description of all sequences. I will not comment on this for the other units. For delta 5.2. I do not see details but it seems to be a forced regression and not a real stillstand.

P10, Line 21 and following: How do you explain the pronounced step in the morphology of the upper boundary of Sequence 3/4?

P11, Line 8: Sequence 3 may fill erosional features but the lower boundary is not erosive. It is the upper boundary of the underlying sequence. Check also for other sequences.

P11, Line 26: This is a correct description (It lies above an erosional unconformity and sediments fill the channels).

P12, Line 3: I assume it should be Sequence 1 (and not 2)

Discussion P13, L3: See general comment concerning lake-level stillstands.

P13, Line 28, Boom gorge has not been introduced before. Refer to Fig. 1, where it is shown.

P15, .Line 10: See previous comment about the anticlines (P7, Line3).

P15, Line 25: What do you mean with 'May have influenced' Again, no detailed information about the anticlines is given in the manuscript. The anticlines are not critical for the manuscript but you draw conclusions based on the anticlines without a real presentation of these anticlines.

P16, L1-5: see general comments. Should be illustrated in a figure.

[Figure]

P16, Line 22 – 28: This has already been partly discussed in the previous section but it also partly contradicts the previous section, where it is stated that subaerially exposed terraces may indicate lake levels 100 m higher than present. Clarify.

P18, Line 3, 4: Is there a reason that you are nor listing rainfall/direct precipitation?

Page 18, Line 15: See general comments about lake-level stillstands.

Conclusions

P19, Line 16: Change to 'each stratigraphic section contains at least 2 . . .'

Figures:

Fig. 1: An overview map showing the general location of the lake and regional features is missing. Colour code would be useful. The profiles shown in the manuscript should be marked much clearer (direct reference to the Figure).

Fig. 2: Depth below lake floor scale is a bit confusing. How have you set the zero point? I would recommend changing the scale to depth beneath present lake level.

Fig. 3: OK. If you show a profile from the western shelf, you should mark some of the prominent deltas identified on both profiles.

Figs. 4-7: See comments above. You need to mark all deltas identified on this figure. Much more delta features than marked on the figure are listed in Table 1 for this profile. There is no reference to Fig. 4 in the text.

Fig. 8: OK
* * *
[Figure]

Clim. Past Discuss.,
doi:10.5194/cp-2016-3-RC2, 2016

[Figure]

**CPD**
This is a well presented and scientifically significant study that is very appropriate to be published in the 'Climate of the Past' journal. The interpretations are sound, well-based on data and provide new insights into a highly dynamic paleoclimate regime in Central Asia. Eventhough the data do unfortunately not allow a dating of the presented wet-dry climate cycles ('a reason to drill the lake'), the presentation of these patterns nevertheless provide novel data that are absolutely worth to be reported.

My main concern refers to limited amount of data shown as figures: The figures (beau-

tifully crafted by the way, a pleasure to look at) focus basically on a single seismic line. The line is spectacular indeed, the seismic stratigraphic interpretation sound and somehow textbook-style, but that same seismic line is shown on 4 full pages with different levels of annotations, way to much. There is no need to show every infilling step of each seismic sequence, the only added value comes in Figs. 6b and 7b, where eroded sequences are reconstructed, but that can also go in a smaller extra figure. What is needed much more are more shown examples. I am curious how representative this singled-out seismic line really is. In fact, many of the discussed delta lobes are not presented but provide crucial elements of the lake-level reconstruction. As reader, I need to see at least 2-3 more examples of seismic lines from other areas of the lake (for instance also the Western delta area), i.e. more of the sequence stratigraphic architecture. This can be done at 'no cost', as Figs. 4-7 can be reduced to one full page, there is plenty of space available. Having said this, I also would appreciate with new figures or maybe also in map view what is really meant with the concept of 'delta lobes' and how they are distributed on both sides of the elongated lake. These lobes, and their vertical and lateral stacking pattern is the key to reconstruct the details of the lake-level curve, so these data are crucial but yet not presented.

I am intrigued by the fact that all sequences and their boundaries on the shown seismic line display a gentle basinward dip. Is this a pattern on all seismic sections, also in the West? Or is this formed by a general forced regressional pattern with falling lake level upon delta progradation? But why is there never an still stand (horizontal progradation) or even an aggradation of a delta sequence upon a gently rising lake level? Is this a function of tectonic subsidence or tilting?

I am also wondering why sequence 5 is not subdivided in two main sequences (currently called subsequences 5.1 and 5.2), as they are separated by a very clear unconformity. What defines the hierarchy of the sequences? On contrast, I am not fully convinced that sequences 2.1 and 2.2 represent clearly two pulses or whether they form a transitional package without major unconformity in-between them. Both of these issues

are hard to track, as one shown seismic line alone from the shelf is not sufficient.

Discussion on p.14 about deltas 1.1-1.6 is hard to follow. I cannot judge on he basis of the limited shown data whether 1.1-1.6 is indeed in chronologic order or whether lake level plays as a 'jojo' reshuffling the lobes in maybe a different order? Moreover, the arguments presented for an uplifted nature of the subaerially exposed terraces are a bit weak, I am somehow not convinced in this matter.

Further comments

The English language can be improved in some of the sections....maybe have an English native speaker go through it.

Shorten title by deleting 'An extended history of', just start with 'High-amplitude lake-level fluctuations of....'

1/21: delete one of the two 'past'

1/22: ....from the Mongolian steppe blocking the mid-latitude Westerly's.

2/6: ... AND thermal expansion...

2/6: HenCe (spell checker!!)

2/6: no comma after curve

2/10: Three 'large' on one line, too much!

2/25: The quoted publication (Anselmetti et al., 2006) initially stated indeed glacial/interglacial cycles, which after drilling turned out to be stadial/interstadials, I would change to:....were correlated to wet-dry paleoclimate patterns with lowstands during the stadials and highstands during the interstadial periods (Hodell et al., 2008, Quaternary Science Reviews 27, 1152-1165)

3/2: Lake-level changes as large as 170 m have on one hand been attributed to....

3/5: Awkward short sentences, change to: The impact crater of Lake Bosumtwi

(Ghana) is....

3/8: ...is purely driven by the evaporation/precipitation ratio.....

3/9: Lake Issyk-Kul, subject of this study, is....

3/21: Figure 1 shows these mountains exactly reversed (N vs. S). Which one is correct?

4/4:' ...and by steep...' poor English, unclear what is meant, reword or make 2 sentences

4/7: can this 110 m depth transition be marked on Fig. 1

4/21: thRough

4/22: Surface-water temperature...

4/26: This is a hydrologically 'bold' statement....any references?

5/16: 2004 is not 'recently'

6/14: avoid one-sentence paragraphs.

Fig. 2: I note a somehow prominent change in basinal sediment geometry (draping vs. filling) at ∼1.1 s twt in the middle of the profile, in particular when correlating to the right side of the figure...is this worth to be discussed?

Make sure final Fig. 2 has sufficient resolution, I have problems seeing for instance the mass-transport deposit.

7/2: became (Spell checker!!)

7/4: Which anticline? Has no been mentioned before

7/4:...dip angle OF the strata....

7/12: two 'however' within 4 words:-(

7/12: But on the figure I only see ca. 200 m of sediments.....the authors report partly 370 m sediment thickness but no evidence is shown.

7/20-25: The longer anticline still is visible on lake floor, correct? It looks like a dipping anticline (towards SW), is that worth to be mentioned? Are the two anticlines aligned in an 'en echelon' pattern? Are these anticlines really tectonic in nature of simply a draping remnant of an underlying basement high?

8/7 and ff.: Use throughout the manuscript 'reflections' instead of 'reflectors'. On seismic data, you only see reflections. Reflectors (=impedance change in the sediment record) cannot have amplitudes.

Seismic facies 3, here the term 'retrogradation' may also be used, or a 'backstepping' delta.

13/6: ...riseS...

13/9:..mainly in the shallower parts of the lake

14/7: this 'some' here and in numerous other places in the text is not elegant: use 'ca.' or even better a '∼'.

14/25: I don't agree with the mentioning of the outflow here: The balance is made by precipitation/inflow and evaporation only (maybe subsurface outflows). The outflow is a result of positive hydrologic balance, thus the difference in the balance, then the lake level is geomorphologically fixed and the system open. If the balance is negative, then the outflow will be zero and the basin closed.

15/10: No one-sentence paragraph

15/20ff: Why don't the authors call the system a half-graben? with the main border fault in the south? It has all indications, correct?

Time constrains on 15/20: suddenly the term 1 Ma pops up? What is the origin? based upon? No age data at all has been presented before!

[Figure]

Discussion on highstands is necessary but what about tectonics as regulator of outflow level? 16/25 ff: It also could suggest that the outflowing area subsided relative to the lake, lowering the topographic outflow point.

The last figure and the general lake-level reconstruction based on 'shallow' sedimentary sequences is highly reminiscent to another study in a Patagonian hydrologically closed lake where the first- and one co-author were also co-authoring: I would also quote this study, as some of the concepts match very nicely (Anselmetti et al., 2009, Sedimentology 56, 873-892)

One should remove the thick red and blue arrows on last figure and make lake-level lines thicker, that will be much better to visualize these impressive lake-level variations,
* * *
[Figure]

Clim. Past Discuss.,
doi:10.5194/cp-2016-3-RC3, 2016

[Figure]

**Climate**

**of the Past**

Discussions

Dear author, I have completed my review of "An extended history of high-amplitude lake-level changes in tectonically active Lake Issyk-Kul (Kyrgyzstan), as revealed by high-resolution seismic reflection data" by Gebhardt et al. The manuscript presents high-resolution sparker profiles from Lake Issyk-Kul and has the potential for being a broad and useful study. I personally enjoyed reading it and highly recommend for its publication. Though, it will need to undergo major revisions before it is acceptable for publication. In my review, I outline both major critiques and minor points in the lists below.

[Figure]

Major Points:

-I strongly suggest the authors to present more seismic profiles from different parts of the lake; this is indeed lacking in the current manuscript. In particular, profiles showing deltas from the western margin would be great in order to compare their internal/external structure with the ones from eastern part of the lake.

-Significant lacking of citations in the results part. The authors, most of the times, do not cite or refer figures in the text. Sometimes, the figures are not large enough to see points menntioned in the text, for instance erosional boundaries, delta lobes. Hence, as a reader it is rather difficult to judge the interpretation.

-Would it be possible to correlate stratigraphic boundaries towards the deeper parts of the lake? I can see that deep lake sediments are characterized by alternating high- and low-amplitude seismic reflections which most likely reflect transgression and regression periods.

-I am also missing isopach or isochron maps of seismic units in order to understand their thickness variations and thus the source regions through lake evolution. If this is not applicable or doable, it is better to mention the average thickness of individual units and possible source regions in the text.

- I suggest the authors to make a new basemap and draw lakeward boundary of the deltas (color-coded) in order to see their lateral extent along the western and eastern shelves. The distribution of sublacustrine channels can also be superimposed on this map.

-The authors presents and discuss structural setting of the lake, however I do not see any structural map showing faults, anticlines, or synclines throughout the lake as well as its surroundings. I see several seismic profiles crossing the anticline structures on the base map but neither of them is shown. It is worth to discuss the relative timing of these structures based on thickness variations of overlying/underlying sediments. Also

a normal fault in the southern part of Profile ik01 (Fig. 2) should be shown.

-I suggest the authors make schematic diagrams (with scale) from East to West showing the formation of deltas throughout the lake formation. The former lake levels should be indicated. This would definitely improve the quality of the manuscript.

Line points:

-Page 6, Line 24. "..presence of a series of faults.." It would be better to show these faults on a map.

-Page 6, Line 25-26. Please locate the "Main Terskey Fault (MTF)" on a map.

-Page 7, Line 2. Change "becaoem" to "became " and "Miocene" to "Miocene"

-Page 7, Line 12. Modify so that it reads, "..However, it is quite likely that . . ."

-Page 7, Line 18-19. "In the southeastern part of the lake, the strata are not inclined as would be expected in this asymmetric basin". Please refer to figure or show a seismic section. I can see that there are various seismic profiles traversing these anticline structures.

Page 7, Line 23-24. "Both anticlines are progressively buried by younger sediments, and the southern one is meanwhile completely leveled by sediments." Please show a seismic section as I cannot confirm whether they are buried or leveled by sediments.

Page 8, Section 4.2 Facies Types. I suggest changing "Facies types" as "Seismic Facies" and the "Facies I" as "Seismic Facies 1 (SF1)". It is easier for descriptions. It is also better to formulate as "SF1 is characterized by. . . " than "this facies type is characterized. . ."

Page 8, Line 11. Clinoforms should be better illustrated; topsets-foresets transitions (if they exist) are not noticeable on the presented seismic profiles. I propose to the authors to add a figure as an example of interpreted delta (for instance, immediately below Fig. 3a; (3b, interpreted section of 3a) in which reflections of topset, foreset and

bottomset are pointed.

Page 9, Lines 1-2. "Facies IV may be interpreted as former delta sediments that have been affected by post-depositional processes (e.g. sediment remobilization, slumping, liquefaction) that caused them to loose their internal structure". They are indeed former delta deposits but I am not sure such reflection configuration was caused by slumping etc. Looking at reflections within Sequences 3/4 in Fig. 4, as a whole package, I do not think it has something to do with slump deposits. Would it be possible that such reflections were due to coarse-grained sediments resulted from rapid loading of rivers?

Page 9, Line 18-19. "The topset-foreset roll-over point is considered as a proxy for the lake level at the time of its forming". Please give a reference.

Page 9, Line 23. Modify so that it reads, "Sequence 7 (S7) is the..." In the following parts you can shorten its name as "S7" instead of "Sequence 7".

Page 9, Line 16. Please delete "lacustrine"

Page 10, Line 2. "..Some of these occur only in the western delta area (7.5, 7.4)" Please refer or show a seismic section.

Page 10, Line 7. "Sequence 6 is clearly visible both on the western and eastern delta areas." Please refer to Fig.

Page 10 , Line 8. "... rather thin..." How much?

Page 10, Line 9. ".... delta lobes could be identified at 461 (no. 6.1) and 361 m bll (6.2)." Could you please label these delta lobes in the seismic sections?

Page 10, Line 13. "Sequence 5 is overlaying sequence 6 with an erosional boundary in between (Fig. xx??).

Page 10, Line 14-15. "The bathymetrically higher delta 5.1 exhibits extensive erosion (Fig. 6b)". I am looking at this figure and it is almost impossible to see the erosional surface. I suggest the authors to show close-up sections to show these features.

[Figure]

Page 10, Line 21. Modify so that it reads, "Sequence 4 overlies . . ."

Page 10, Line 24-25. Modify so that it reads, "In Sequence 4, the delta lobes are characterized by predominantly SF4, but. . ."

Page 10, Line 26. Change "well-layered" into "well-stratified". Please also make the colors of sequences more transparent so that the internal reflections can be seen clearly.

Page 11, Lines 1-2. "Three delta lobes were identified: the oldest (4.1) at ca. 319 m bll, followed by a delta (4.2) at approximately 250 m bll and a third, younger (4.3), at 397 m bll." Where are they in the seismic section? Please mark the locations of these deltas.

Page 11, Line 6. " Sequence 3 could only be clearly identified in the western delta areas;. . ." Please show a seismic profile from the western area which clearly depicts S3.

Page 11, Lines 8-9. "In the western delta complex, Sequence 3 is characterized by a lower boundary that was partially erosive into the underlying sediments but grades into a correlative conformity in other places." I cannot judge this interpretation as I do not see any figure showing this relationship.

Page 13, Lines 4-5. Instead of using lake level decrease and increase, how about using regression and transgression?

Page 14, Line 3. Change "Subaquatic channels" into " Sublacustrine channels"

Page 15, Line 18. ". . .subsidence seems to have been relatively constant through time." Can you quantify the fault activity by looking at thickness variations towards it?

Figure Captions Overall, the figure captions should be improved.

Figures

Fig. 1. Please add an inset map showing large areas of the regions. With the current

[Figure]

map, I cannot say where the Lake Issyk-Kul is located. The depth color bar is missing as well. What about the bathymetry of the lake reconstructed from seismic reflection profiles?

Fig. 2. I suggest including vertical exaggeration for all seismic profiles. Locate the fault on the southern end of the profile. Can you please enlarge the MTDs?

Fig. 3. Please add vertical and horizontal scales.

Fig. 4. It would also be better to give names for the sequence boundaries, such as Sequence boundary 1 (SB1) to SB7. But it is your choice.

Fig. 5. Please switch the Figure 5a and 5b. It should be displayed in an order and should start from Sequence 7. Please do this for the following figures.

Regards

[Figure]

Dear Editor,

I hereby comment the reviewer's comments on my manuscript and outline the changes that were made to the manuscript before resubmission.

**Review #1**

1) Show more data and give a better general description of the data...
First of all, I added another profile from the western delta to show how similar the profiles are from both ends of the lake. All deltas visible on the two profiles were labeled. Second, description of seismic facies types was improved, and a chapter containing visual description of the three profiles shown was added. Third, a paragraph as well as a figure on the detection of the sequence boundary was added for better understanding how this study was carried out. Fourth, the description of the sequences was kept in the text, but improved, and supported by the points mentioned before, this has lead to a significantly improved description of data.

2) Usage of terminology
Usage of terminology was fully checked and improved throughout the manuscript.
3) Clinoforms deposited not only during stillstands, but also during times of relatively constant lake level
Yes, I agree. This was mentioned in the text.
4) Distribution of delta sequences
In the two profiles from the western and eastern deltas, respectively, all deltas encountered were labeled in consistency with table 1. I also added thoughts on differential sediment supply leading to formation or non-formation of deltas to the manuscript.
5) General map
The regional map in fig. 1 was improved, and an inset showing the location of the lake in its larger setting was added. Geographical names used in the text were added.
6) Outlook
I added a small outlook in the conclusions, stating that only drilling could solve the problem of dating sequences.

Specific comments: All comments were included/text and figures were changed accordingly except of:
P6, Line 22: Indeed, I do not show seismic data down to 375 m. It is not worth changing the seismic figures (information in greater depths as shown in the current figures is sparse, hence showing this part of the profile would only make the resolution of the profiles worse without adding much information). I deleted this statement, it is not crucial for the interpretation.
P7, Line 3; P7, Lines 19/20; P15, Line 10; P15, Line 25: The anticline structure is not crucial for interpretation of lake-level changes in this lake. All tectonic features are subject of a second paper by a student that is currently under work. In order not to jeopardize the student's work, I did not deepen unnecessarily the discussion on tectonic features but just deleted these parts of the text.

**Review #2:**

Similar to Review #1, additional seismic data is requested by deleting some of the old figures instead. I added one profile from the western delta and labeled all deltas, additionally a map of sequence thickness was added for better visualization of the delta structure described in the text. I did not add a figure explaining the concepts of delta reconstruction – this was not requested by the other reviewers either – but improved the delta zoom-in in the figure on seismic facies, where topsets, foresets and bottomsets were labeled.

All seismic profiles indeed show a gentle basinward dip, similarly on both the western and eastern region. This is now more clearly visible from the second seismic profile that was shown

in the manuscript. The reason for this, however, remains unclear. Tectonic subsidence or tilting seems to take place in a north-south direction (shown and discussed on profile ik01) but is unlikely in west-east direction, as profiles from both sides dip towards the lake center.

Sequence 5 was not subdivided into two main sequences as it was interpreted as representing falling lake level. Only where the lower delta was formed, the topsets form onlaps onto the lower strata within the sequence. Considering the newly shown profile from the western delta we think that this becomes clearer. Also for deltas 2.2 and 2.1 it should now be more obvious that these are two different pulses.

I agree that from the previously shown profile it remains unclear if the deltas 1.1 to 1.6 are single deltas. With the new profile it becomes clearer that at least deltas 1.6 and 1.3 were independently built up. Discussion on the subaerially exposed terraces was improved and corrected. In the previous version the 30 m of water depth above the currently formed delta was added to the terrace heights, which was simply wrong. The exposed terraces are strand terraces, not delta lobes. This should now be much more understandable.

Further comments:
All further smaller comments were included and text was changed accordingly. The quote "Anselmetti et al 2006" was replaced and the text adapted. The two mountain chains were mentioned in wrong order in the manuscript before, this was changed. Anything on the anticline was deleted from the text, see comment to review #1. Time constrains on 15/20: The term 1 Ma was not changed in a previous version; it should have been the 800 ka that were estimated from sedimentation rates. This value however is absolutely vage and not needed in this context – it was simply deleted. Laguna Potrok Aike in Patagonia was added to the introduction, including a reference to Anselmetti et al. 2009 as requested.
Not included/changed:
Fig. 2: prominent change in basinal geology at a depth of 1.1 s in profile ik01 – this is not crucial for the current manuscript on lake-level change, but will be discussed in the student's tectonic paper.
14/25: I do not agree that the outflow should not be mentioned here. The lake is located in a tectonic setting, hence a sudden change in outflow by blocking (e.g. by a landslide or another tectonic event) has a direct influence on the hydrologic balance, which is not only driven by E/P but also directly influenced by a sudden change in water volume.
15/20ff: My co-author Ed Sobel is carrying out tectonic work in this region. He did not agree with the term halfgraben for the lake basin that I used in a former version of this manuscript. This term will hence not be used in the revised version.

**Review #2:**

Major points:
An additional profile from the western part of the lake is now shown, interpreted and labeled similar to the one from the eastern profile. Seismic facies description is also improved, and seismic details allow now to see how sequence boundaries were depicted.
Figures are now more often cited in the results part.
Correlation of stratigraphic boundaries towards the deeper part of the lake is impossible. On one hand, most profiles stop before they reach the lake floor (both figures 5 and 6 show the entire length of the profile). On the other hand, seismic reflections thin out towards the foot of the slope, making it impossible to follow them down into the deeper parts of the lake. It is also likely that the more central part of the lake is strongly dominated by turbidites. The characteristic alternating high- and low-amplitude reflections therefore more likely represent turbidite sequences than the transgressional and regressional periods.
I added a map of sequence thickness, but discussed my concerns about direct interpretation as already outlined in my answer during the open review process. Fig. 8 shows clearly that large

parts of the initial sequences were eroded, and depot centers during deposition are not necessarily reflected by thick sediment packages. In turn, apparent depot centers in the thickness map are not necessarily reflecting the delta lobes, but likely also infill of erosional features into the underlying sequence. In addition, spacing of the seismic lines is in the order of 1-2 km, and in a highly dynamic system such as a delta, this spacing is too big to reliably generate maps (and hence cell size during gridding had to be chosen at 500 m to get reasonable spatial coverage). Having said this, I did also not add a map showing the lakeward boundaries of the deltas.

Structural/tectonic information that was not crucial for discussion of lake-level changes was removed from the manuscript and will be presented in more detail in a subsequent manuscript currently prepared by a student.

Schematic diagram from east to west – I tried several times to draw this but was not successful. I would have liked to show lake evolution as nicely as e.g. in Cukur et al., 2014, their fig. 8 (Cukur et al., 2014: Water level changes in Lake Van, Turkey, during the past ca. 600 ka: climatic, volcanic, and tectonic controls. J Paleolimnol 52:201-214). Unfortunately, in our case (a) we do not have profiles from the eastern delta that reach down to the lake floor.  (b) We do not have profiles that cross the lake from the western to the eastern delta. And (c) sequences 7 to 2 were eroded significantly. Additionally, timing of the subaerially exposed terraces that likely form the uppermost level of one of the sequences is unclear. We also do not know if e.g. a higher lake level was eroded (erosional discordances in between sequences). And furthermore profiles do not reach the most proximal part of the lake where another delta lobe is clearly visible from bathymetry, and where older delta lobes may be buried. Drawing such a schematic diagram at present would be highly speculative. This should be postponed until better seismic data and groundtruthing by drilling is available.

Minor points:

All suggestions were included/text and figures changed accordingly. Sequence names were shortened to S1 to S7, and also seismic facies types were changed to SF 1 to SF 4. Deltas in the profiles shown were labeled. Vertical exaggeration was calculated for all seismic profiles and included. Vertical and horizontal scales were added to the figure with the seismic facies types. Color bar was added to fig. 1, and reference to the digital elevation model is given. The only comments that were not included where those on tectonic content, e.g. the anticline structure, see comments to review #1.

**List of major changes:**
- All figures were improved, a seismic profile from the western delta was added and some redundant figures from the eastern delta were removed
- A figure showing sequence thicknesses was added as well
- Figure 1 now shows the geographic names mentioned in the text
- Seismic profiles are labeled with all deltas visible, in accordance with table 1
- The Results & Interpretation chapter was partly rewritten and reorganized: (a) seismic facies analysis was rewritten and improved. (b) a small subchapter on the identification of stratigraphic sequences was added. (c) description of the seismic profiles was added. (d) description and interpretation of stratigraphic sequences was improved.

The comments by the three anonymous reviewers were very helpful and significantly increased the quality of the manuscript. I would be pleased if you could consider the manuscript in its revised form for publication.

Best regards,

Catalina Gebhardt

[revised manuscript text omitted]

Catalina Gebhardt 8.7.16 00:10

Catalina Gebhardt 8.7.16 00:10

Catalina Gebhardt 8.7.16 00:10

Catalina Gebhardt 8.7.16 00:10

Catalina Gebhardt 8.7.16 00:10

Catalina Gebhardt 8.7.16 00:10

Catalina Gebhardt 8.7.16 00:10

Catalina Gebhardt 8.7.16 00:10

Catalina Gebhardt 8.7.16 00:10

Catalina Gebhardt 8.7.16 00:10

Catalina Gebhardt 8.7.16 00:10

Catalina Gebhardt 8.7.16 00:10

Catalina Gebhardt 8.7.16 00:10

Catalina Gebhardt 8.7.16 00:10

Catalina Gebhardt 8.7.16 00:10

Catalina Gebhardt 8.7.16 00:10

Catalina Gebhardt 8.7.16 00:10

Catalina Gebhardt 8.7.16 00:10

Catalina Gebhardt 8.7.16 00:10

[revised manuscript text omitted]

---

## Author Response (AR2)

Dear Editor,

I hereby comment the reviewer's comments on my manuscript and outline the changes that were made to the manuscript before 2nd resubmission.

**Report #1**

p.3/l. 2: Lake XY is located in Country Z, and all other sentences with '...is located....': I would not make a single-sentence statement here for the country, just put the country in () after you mention the lake. But it is politically not correct to place Lake Lisan entirely in Israel; there is the Jordan and a Palestine part of modern Dead Sea, thus also of the higher-level Lake Lisan - may be just say '...in the Levant...'

*Where appropriate, sentences were changed. Where more geographical information than just the country name was given, I did not change sentences. Israel was changed into Levant.*

p.5/l.14: 'uplifted terraces' could also be caused by higher lake level! This feeds in discussion on early p.15, where terraces are only discussed in terms of lake level and not anymore of tectonics! It is finally mentioned on p.15/l.19 (uplifted) p.16/l.12, but I would open this section and stating the two endmember options (hydrology-induced lake-level changes vs. uplift/subsidences at the beginning of this section.

*I added a sentence on the two endmembers, but not at the beginning of the chapter but at the end where it seemed more appropriate to me.*

p.8 and other parts: I repeat my statement of the initial review: On seismic data, you only see reflections (not reflectors). Reflections are (apparently) truncated, not reflectors.

*I changed reflectors to reflections.*

p.10/l.8: I also would indicate that 1500 m/s is truly a minimal velocity for such sediments: Coarse-grained deposits such as sand and gravel will be significantly faster. One should mention that thus the thicknesses are also from this perspective minimal values.

*Added. We however surely did not encounter gravel; a sparker system would not be able to penetrate through such coarse-grained sediments.*

Fig. 9: The curve with the lake-level reconstruction: if the lake level would evolve in the fashion shown with the red line, it would not form delta sequences. Each delta represents, as stated in the paper elsewhere, a stillstand or slowdown of lake level. I suggest that is schematically shown in Figure 9 by making the red line step-shaped, i.e. between subsequences of 7 (falling) or all subsequences of 1 (rising).

*I changed figure 9 accordingly.*

p. 14/l.15-20: I also would mention the increased compaction in deeply buried sediments here, that will reduce the apparent sedimentation rate (apparent in contrast to mass-accumulation rate).

*Done.*

p.20/l.23-25: These mentioned sequence boundaries do not represent times of strong or weak Siberian highs, but they represent times of shifts: The high is strong or weak within the sequence with overall changed lake level, but the boundary indicates onset if changes, establishment of a new depositional sequence due to new climate regime.

*This is true, and the text was changed accordingly.*

**Report #2**

1) Page 6, Line 10: You mention that it is not possible to trace sequence boundaries between the eastern and western delta complexes but it was still possible to identify the same sequences in both areas due to similar TWT depth. This is a weak argument as tilting may have occurred due to tectonic activity. Can you provide other arguments? Maybe the succession of sedimentary units with similar characteristics identified in both areas supports this statement and should be mentioned. Perhaps you have additional arguments for this statement.

*I agree that at first glance this argument might seem weak.*
*On the other hand, tilting in this lake occurs in N-S direction, not in W-E direction, as can be seen on fig. 4, and the identification of delta lobes at similar depths on both the eastern and western delta – which was mostly possible in more than one profile each - confirms that the lake has not experienced different tectonic dynamics between E and W.*
*P. 6 line 10 is only the methods section where I state which approach was used, justification and discussion should not be done in this paragraph. I think that it was clearly demonstrated throughout the entire manucript that this approach using delta lobe depths is justified.*
*The succession of sedimentary units or layers is not unique enough to use it for identification on both delta complexes. Deltas are unfortunately highly variable on a local scale and would not allow this approach. Furthermore, erosion at the stratigraphic boundaries is unequally pronounced, further hindering this approach.*
*The paragraph was hence not altered, this is the approach we used.*

2) Page 8, Line 5: I do not understand this sentence: Erosion truncates older strata hence form toplaps or onlaps onto the erosional discordance. What is eroded and what form toplaps and onlaps? Toplap is a termination against an upper boundary caused by non-deposition. Hence, they cannot be toplap onto the erosional surface. Check also caption for Fig. 3 where you have the identical sentence. I cannot follow your nomenclature in Fig. 3b and 3c. Some of your downlaps seem to be onlaps (blue lines on Fig. 3b). The red labels are onlaps and toplaps but toplap is against an upper boundary and onlap against a lower boundary. Check and clarify.

*Labels in 3b were indeed mis-labelled in one case. This was changed.*

*I do however not completely agree that no erosion can be involved when using the term „toplap". Using the SEPM STRATA definition some minor erosion can well be involved:*
***„Termination of strata against an overlying surface mainly as a result of nondposition (sedimentary bypassing) with perhaps only minor erosion."***

*Toplap is the only term to name for strata that lap out at the upper boundary, all other terms are used for lapouts towards the lower boundary. Hence, this term was not changed.*

3) Mark the profiles shown in the manuscript much clearer on Fig. 1

*Done. I also changed the figure caption accordingly.*

4) You show seismic line 49 on Fig. 3 and 5. Is it possible to combine these two figures?

*Both figures are rather complex and do not exactly focus on the same thing. I would like to avoid combining these figures; a combined figure would be too complex to be easily comprehensible. Furthermore, one of first round's the reviewers requested more information on how toplaps/downlaps/onlaps etc. were identified.*

5) Page 9, Line 20. You mention that the sequence boundary between S3 and S4 could not be identified. What does it mean? Is this unit comprising both S3 and S4 or only one of these units? I understand your text in that way, that S5 is the same sequence on both profiles (Fig. 5 and 6) but you used different colours for Unit S5 on the Figs. 5 and 6. Use same colours. Also use similar colours for S3 and S4 on Fig. 5, which is only one Unit on Fig. 6.

*I changed the color of S5 in fig. 6 to match S5 in fig. 5. I however would like to keep two different colors for S3 and S4 in fig. 5 because they are separated by a sequence boundary, and only one color in fig. 6 because either only S4 is visible here, or the sequence boundary cannot be identified. I however changed the combined „S3&S4" color to that of S4 in fig. 6, because S4 is surely part of „S3&S4".*

6) Page 9, Line 24: Should be issyk049 not issyk0149.

*Yes! Changed, thanks.*

7) My impression is that you see delta sediments in S6 on the most western part of Fig. 6, which are not marked on the figure. Do you agree? If yes, please mark them.

*It is true that a highly eroded remainder of a delta lobe can be spotted in S6 on Fig. 6. I however cannot identify its true depth and can hence also not identify to which of the terraces identified in other profiles it may correspond. It was not considered in the calculations. I named this delta lobe „6.x" and added text to the figure caption.*

8) Page 11, Line 25: lines 26 and 27: Repetition from the previous paragraph.

*I changed wording.*

9) Line 12, Line 7 and following: You state that S3 is deposited during a lake level rise. The boundary to S2 may be erosive in parts but seem to be non-erosive in other parts. In the previous section you state that S3 cannot be distinguished from S3 or it is completely eroded. If S3 is deposited during a rising lake level and you do not have very clear signs of erosion at the boundary between S2 and S3, I cannot see, how S3 should have been eroded in the eastern part of the lake. In addition, a few lines up you state that

S3 is mostly transparent in the eastern part. This would imply that S3 is indeed deposited which is questioned in the previous paragraph. Clarify.

*I agree that the paragraph dealing with S3, S4, and S3&S4 was not written accurately enough. I changed from S3 to „S3&S4" where I write about the layer between S5 and S2 on the eastern delta and changed the text to clarify that we are not sure if S3 is still there on the eastern delta system or if it was completely eroded.*

10) Page 17, Line 16: The anticlinal structures are not mentioned in the text before but only shown on Fig. 1. This is a bit confusing. I am aware that you decided to delete the part with the anticlinal structure but if you need them for the discussion, you have to introduce them somewhere. I think you can delete the remark on the anticlinal structures. Just mention that similar depths of the deltas in the western and eastern part do not support the idea that both areas experienced significantly different histories.

*Thanks, I did not notice that there were still some remains in the text mentioning the anticline structure. This was deleted from the text as suggested.*

11) Page 21, Line 10: Not really still stands but times of relatively constant lake levels.

*Changed.*

The comments by the two anonymous reviewers were very helpful to finalize the manuscript. I would be pleased if you could consider the manuscript in its revised form for publication.

Best regards,

Catalina Gebhardt

[revised manuscript text omitted]